# Convolutional State Space Models for Long-Range Spatiotemporal Modeling

**Jimmy T.H. Smith**[*,2,4], **Shalini De Mello**[1], **Jan Kautz**[1], **Scott W. Linderman**[3,4], **Wonmin Byeon**[1]

[1]NVIDIA, [*]Work performed during internship at NVIDIA
[2]Institute for Computational and Mathematical Engineering, Stanford University.
[3]Department of Statistics, Stanford University.
[4]Wu Tsai Neurosciences Institute, Stanford University.
{jsmith14,scott.linderman}@stanford.edu
{shalinig,jkautz,wbyeon}@nvidia.com.

## Abstract

Effectively modeling long spatiotemporal sequences is challenging due to the need to model complex spatial correlations and long-range temporal dependencies simultaneously. ConvLSTMs attempt to address this by updating tensor-valued states with recurrent neural networks, but their sequential computation makes them slow to train. In contrast, Transformers can process an entire spatiotemporal sequence, compressed into tokens, in parallel. However, the cost of attention scales quadratically in length, limiting their scalability to longer sequences. Here, we address the challenges of prior methods and introduce convolutional state space models (ConvSSM)[1] that combine the tensor modeling ideas of ConvLSTM with the long sequence modeling approaches of state space methods such as S4 and S5. First, we demonstrate how parallel scans can be applied to convolutional recurrences to achieve subquadratic parallelization and fast autoregressive generation. We then establish an equivalence between the dynamics of ConvSSMs and SSMs, which motivates parameterization and initialization strategies for modeling long-range dependencies. The result is ConvS5, an efficient ConvSSM variant for long-range spatiotemporal modeling. ConvS5 significantly outperforms Transformers and ConvLSTM on a long horizon Moving-MNIST experiment while training $3\times$ faster than ConvLSTM and generating samples $400\times$ faster than Transformers. In addition, ConvS5 matches or exceeds the performance of state-of-the-art methods on challenging DMLab, Minecraft and Habitat prediction benchmarks and enables new directions for modeling long spatiotemporal sequences.

## 1 Introduction

Developing methods that efficiently and effectively model long-range spatiotemporal dependencies is a challenging problem in machine learning. Whether predicting future video frames [1, 2], modeling traffic patterns [3, 4], or forecasting weather [5, 6], deep spatiotemporal modeling requires simultaneously capturing local spatial structure and long-range temporal dependencies. Although there has been progress in deep generative modeling of complex spatiotemporal data [7–12], most prior work has only considered short sequences of 20-50 timesteps due to the costs of processing long spatiotemporal sequences. Recent work has begun considering sequences of hundreds to thousands of timesteps [13–16]. As hardware and data collection of long spatiotemporal sequences continue to improve, new modeling approaches are required that scale efficiently with sequence length and effectively capture long-range dependencies.

---

[1]Implementation available at: https://github.com/NVlabs/ConvSSM.

37th Conference on Neural Information Processing Systems (NeurIPS 2023).

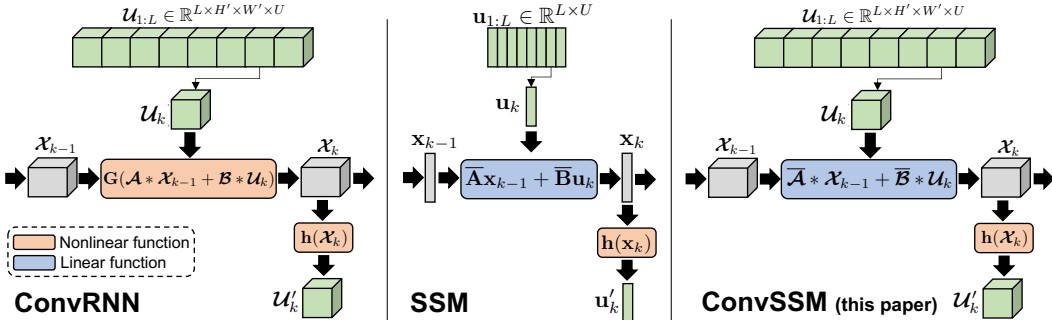

Figure 1: ConvRNNs [17, 18] (left) model spatiotemporal sequences using tensor-valued states, $\mathcal{X}_k$, and a nonlinear RNN update, $\mathbf{G}()$, that uses convolutions instead of matrix-vector multiplications. A position-wise nonlinear function, $\mathbf{h}()$, transforms the states into the output sequence. Deep SSMs [19, 20] (center) model vector-valued input sequences using a discretized linear SSM. The linear dynamics can be exploited to parallelize computations across the sequence and capture long-range dependencies. We introduce ConvSSMs (right) that model spatiotemporal data using tensor states, like ConvRNNs, and linear dynamics, like SSMs. We also introduce an efficient ConvSSM variant, ConvS5, that can be parallelized across the sequence with parallel scans, has fast autoregressive generation, and captures long-range dependencies.

Convolutional recurrent networks (ConvRNNs) such as ConvLSTM [17] and ConvGRU [18] are common approaches for spatiotemporal modeling. These methods encode the spatial information using tensor-structured states. The states are updated with recurrent neural network (RNN) equations that use convolutions instead of the matrix-vector multiplications in standard RNNs (e.g., LSTM/-GRUs [21, 22]). This approach allows the RNN states to reflect the spatial structure of the data while simultaneously capturing temporal dynamics. ConvRNNs inherit both the benefits and the weaknesses of RNNs: they allow fast, stateful autoregressive generation and an unbounded context window, but they are slow to train due to their inherently sequential structure and can suffer from the vanishing/exploding gradient problem [23].

Transformer-based methods [9, 13, 14, 24–27] operate on an entire sequence in parallel, avoiding these training challenges. Transformers typically require sophisticated compression schemes [28–30] to reduce the spatiotemporal sequence into tokens. Moreover, Transformers use an attention mechanism that has a bounded context window and whose computational complexity scales quadratically in sequence length for training and inference [31, 32]. More efficient Transformer methods improve on the complexity of attention [33–39], but these methods can fail on sequences with long-range dependencies [40, 13]. Some approaches combine Transformers with specialized training frameworks to address the attention costs [13]. However, recent work in deep state space models (SSMs) [19, 41, 42, 20, 43], like S4 [19] and S5 [20], has sought to overcome attention's quadratic complexity while maintaining the parallelizability and performance of attention and the statefulness of RNNs. These SSM layers have proven to be effective in various domains such as speech [44], images [45] and video classification [45, 46]; reinforcement learning [47, 48]; forecasting [49] and language modeling [50–53].

Inspired by modeling ideas from ConvRNNs and SSMs, we introduce *convolutional state space models* (ConvSSMs), which have a tensor-structured state like ConvRNNs but a continuous-time parameterization and linear state updates like SSM layers. See Figure 1. However, there are challenges to make this approach scalable and effective for modeling long-range spatiotemporal data. In this paper, we address these challenges and provide a rigorous framework that ensures both computational efficiency and modeling performance for spatiotemporal sequence modeling. First, we discuss computational efficiency and parallelization of ConvSSMs across the sequence for scalable training and fast inference. We show how to parallelize linear convolutional recurrences using a binary associative operator and demonstrate how this can be exploited to use parallel scans for subquadratic parallelization across the spatiotemporal sequence. We discuss both theoretical and practical considerations (Section 3.2) required to make this feasible and efficient. Next, we address how to capture long-range spatiotemporal dependencies. We develop a connection between

the dynamics of SSMs and ConvSSMs (Section 3.3) and leverage this, in Section 3.4, to introduce a parameterization and initialization design that can capture long-range spatiotemporal dependencies.

As a result, we introduce *ConvS5*, a new spatiotemporal layer that is an efficient ConvSSM variant. It is parallelizable and overcomes difficulties during training (e.g., vanishing/exploding gradient problems) that traditional ConvRNN approaches experience. ConvS5 does not require compressing frames into tokens and provides an unbounded context. It also provides fast (constant time and memory per step) autoregressive generation compared to Transformers. ConvS5 significantly outperforms Transformers and ConvLSTM on a challenging long horizon Moving-MNIST [54] experiment requiring methods to train on 600 frames and generate up to 1,200 frames. In addition, ConvS5 trains $3\times$ faster than ConvLSTM on this task and generates samples $400\times$ faster than the Transformer. Finally, we show that ConvS5 matches or exceeds the performance of various state-of-the-art methods on challenging DMLab, Minecraft, and Habitat long-range video prediction benchmarks [13].

## 2 Background

This section provides the background necessary for ConvSSMs and ConvS5, introduced in Section 3.

### 2.1 Convolutional Recurrent Networks

Given a sequence of inputs $\mathbf{u}_{1:L} \in \mathbb{R}^{L \times U}$, an RNN updates its state, $\mathbf{x}_k \in \mathbb{R}^P$, using the state update equation $\mathbf{x}_k = \mathbf{F}(\mathbf{x}_{k-1}, \mathbf{u_k})$, where $\mathbf{F}()$ is a nonlinear function. For example, a vanilla RNN can be represented (ignoring the bias term) as

$$\mathbf{x}_k = \tanh(\mathbf{A}\mathbf{x}_{k-1} + \mathbf{B}\mathbf{u_k}) \tag{1}$$

with state matrix $\mathbf{A} \in \mathbb{R}^{P \times P}$, input matrix $\mathbf{B} \in \mathbb{R}^{P \times U}$ and $\tanh()$ applied elementwise. Other RNNs such as LSTM [21] and GRU [22] utilize more intricate formulations of $\mathbf{F}()$.

*Convolutional recurrent neural networks* [17, 18] (ConvRNNs) are designed to model spatiotemporal sequences by replacing the vector-valued states and inputs of traditional RNNs with tensors and substituting matrix-vector multiplications with convolutions. Given a length $L$ sequence of frames, $\boldsymbol{\mathcal{U}}_{1:L} \in \mathbb{R}^{L \times H' \times W' \times U}$, with height $H'$, width $W'$ and $U$ features, a ConvRNN updates its state, $\boldsymbol{\mathcal{X}}_k \in \mathbb{R}^{H \times W \times P}$, with a state update equation $\boldsymbol{\mathcal{X}}_k = \mathbf{G}(\boldsymbol{\mathcal{X}}_{k-1}, \boldsymbol{\mathcal{U}}_k)$, where $\mathbf{G}()$ is a nonlinear function. Analogous to (1), we can express the state update equation for a vanilla ConvRNN as

$$\boldsymbol{\mathcal{X}}_k = \tanh(\boldsymbol{\mathcal{A}} * \boldsymbol{\mathcal{X}}_{k-1} + \boldsymbol{\mathcal{B}} * \boldsymbol{\mathcal{U}}_k), \tag{2}$$

where $*$ is a spatial convolution operator with state kernel $\boldsymbol{\mathcal{A}} \in \mathbb{R}^{P \times P \times k_{\mathcal{A}} \times k_{\mathcal{A}}}$ (using an [output features, input features, kernel height, kernel width] convention), input kernel $\boldsymbol{\mathcal{B}} \in \mathbb{R}^{P \times U \times k_{\mathcal{B}} \times k_{\mathcal{B}}}$ and $\tanh()$ is applied elementwise. More complex updates such as ConvLSTM [17] and ConvGRU [18] are commonly used by making similar changes to the LSTM and GRU equations, respectively.

### 2.2 Deep State Space Models

This section briefly introduces deep SSMs such as S4 [19] and S5 [20] designed for modeling long sequences. The ConvS5 approach we introduce in Section 3 extends these ideas to the spatiotemporal domain.

**Linear State Space Models**  Given a continuous input signal $\mathbf{u}(t) \in \mathbb{R}^U$, a latent state $\mathbf{x}(t) \in \mathbb{R}^P$ and an output signal $\mathbf{y}(t) \in \mathbb{R}^M$, a continuous-time, linear SSM is defined using a differential equation:

$$\mathbf{x}'(t) = \mathbf{A}\mathbf{x}(t) + \mathbf{B}\mathbf{u}(t), \qquad \mathbf{y}(t) = \mathbf{C}\mathbf{x}(t) + \mathbf{D}\mathbf{u}(t), \tag{3}$$

and is parameterized by a state matrix $\mathbf{A} \in \mathbb{R}^{P \times P}$, an input matrix $\mathbf{B} \in \mathbb{R}^{P \times U}$, an output matrix $\mathbf{C} \in \mathbb{R}^{M \times P}$ and a feedthrough matrix $\mathbf{D} \in \mathbb{R}^{M \times U}$. Given a sequence, $\mathbf{u}_{1:L} \in \mathbb{R}^{L \times U}$, the SSM can be discretized to define a discrete-time SSM

$$\mathbf{x}_k = \overline{\mathbf{A}}\mathbf{x}_{k-1} + \overline{\mathbf{B}}\mathbf{u}_k, \qquad \mathbf{y}_k = \mathbf{C}\mathbf{x}_k + \mathbf{D}\mathbf{u}_k, \tag{4}$$

where the discrete-time parameters are a function of the continuous-time parameters and a timescale parameter, $\Delta$. We define $\overline{\mathbf{A}} = \text{DISCRETIZE}_{\text{A}}(\mathbf{A}, \Delta)$ and $\overline{\mathbf{B}} = \text{DISCRETIZE}_{\text{B}}(\mathbf{A}, \mathbf{B}, \Delta)$ where DISCRETIZE() is a discretization method such as Euler, bilinear or zero-order hold [55].

**S4 and S5**  Gu et al. [19] introduced the *structured state space sequence* (S4) layer to efficiently model long sequences. An S4 layer uses many continuous-time linear SSMs, an explicit discretization step with learnable timescale parameters, and position-wise nonlinear activation functions applied to the SSM outputs. Smith et al. [20] showed that with several architecture changes, the approach could be simplified and made more flexible by just using one SSM as in (3) and utilizing parallel scans. SSM layers, such as S4 and S5, take advantage of the fact that linear dynamics can be parallelized with subquadratic complexity in the sequence length. They can also be run sequentially as stateful RNNs for fast autoregressive generation. While a single SSM layer such as S4 or S5 has only linear dynamics, the nonlinear activations applied to the SSM outputs allow representing nonlinear systems by stacking multiple SSM layers [56–58].

**SSM Parameterization and Initialization**  Parameterization and initialization are crucial aspects that allow deep SSMs to capture long-range dependencies more effectively than prior attempts at linear RNNs [59–61]. The general setup includes continuous-time SSM parameters, explicit discretization with learnable timescale parameters, and state matrix initialization using structured matrices inspired by the HiPPO framework [62]. Prior research emphasizes the significance of these choices in achieving high performance on challenging long-range tasks [19, 20, 56, 57]. Recent work [57] has studied these parameterizations/initializations in more detail and provides insight into this setup's favorable initial eigenvalue distributions and normalization effects.

## 2.3  Parallel Scans

We briefly introduce parallel scans, as used by S5, since they are important for parallelizing the ConvS5 method we introduce in Section 3. See Blelloch [63] for a more detailed review. A scan operation, given a binary associative operator $\bullet$ (i.e. $(a \bullet b) \bullet c = a \bullet (b \bullet c)$) and a sequence of $L$ elements $[a_1, a_2, ..., a_L]$, yields the sequence: $[a_1, (a_1 \bullet a_2), ..., (a_1 \bullet a_2 \bullet ... \bullet a_L)]$.

Parallel scans use the fact that associative operators can be computed in any order. A parallel scan can be defined for the linear recurrence of the state update in (4) by forming the initial scan tuples $c_k = (c_{k,a}, c_{k,b}) := (\overline{\mathbf{A}}, \ \overline{\mathbf{B}}\mathbf{u}_k)$ and utilizing a binary associative operator that takes two tuples $q_i, q_j$ (either the initial tuples $c_i, c_j$ or intermediate tuples) and produces a new tuple of the same type, $q_i \bullet q_j := (q_{j,a} \odot q_{i,a}, \ q_{j,a} \otimes q_{i,b} + q_{j,b})$, where $\odot$ is matrix-matrix multiplication and $\otimes$ is matrix-vector multiplication. Given sufficient processors, the parallel scan computes the linear recurrence of (4) in $O(\log L)$ sequential steps (i.e., depth or span) [63].

# 3  Method

This section introduces convolutional state space models (ConvSSMs). We show how ConvSSMs can be parallelized with parallel scans. We then connect the dynamics of ConvSSMs to SSMs to motivate parameterization. Finally, we use these insights to introduce an efficient ConvSSM variant, ConvS5.

## 3.1  Convolutional State Space Models

Consider a continuous tensor-valued input $\boldsymbol{\mathcal{U}}(t) \in \mathbb{R}^{H' \times W' \times U}$ with height $H'$, width $W'$, and number of input features $U$. We will define a continuous-time, linear convolutional state space model (*ConvSSM*) with state $\boldsymbol{\mathcal{X}}(t) \in \mathbb{R}^{H \times W \times P}$, derivative $\boldsymbol{\mathcal{X}}'(t) \in \mathbb{R}^{H \times W \times P}$ and output $\boldsymbol{\mathcal{Y}}(t) \in \mathbb{R}^{H \times W \times U}$, using a differential equation:

$$\boldsymbol{\mathcal{X}}'(t) = \boldsymbol{\mathcal{A}} * \boldsymbol{\mathcal{X}}(t) + \boldsymbol{\mathcal{B}} * \boldsymbol{\mathcal{U}}(t) \tag{5}$$

$$\boldsymbol{\mathcal{Y}}(t) = \boldsymbol{\mathcal{C}} * \boldsymbol{\mathcal{X}}(t) + \boldsymbol{\mathcal{D}} * \boldsymbol{\mathcal{U}}(t) \tag{6}$$

where $*$ denotes the convolution operator, $\boldsymbol{\mathcal{A}} \in \mathbb{R}^{P \times P \times k_A \times k_A}$ is the state kernel, $\boldsymbol{\mathcal{B}} \in \mathbb{R}^{P \times U \times k_B \times k_B}$ is the input kernel, $\boldsymbol{\mathcal{C}} \in \mathbb{R}^{U \times P \times k_C \times k_C}$ is the output kernel, and $\boldsymbol{\mathcal{D}} \in \mathbb{R}^{U \times U \times k_D \times k_D}$ is the feedthrough kernel. For simplicity, we pad the convolution to ensure the same spatial resolution, $H \times W$, is maintained in the states and outputs. Similarly, given a sequence of $L$ inputs, $\boldsymbol{\mathcal{U}}_{1:L} \in \mathbb{R}^{L \times H' \times W' \times U}$, we define a discrete-time convolutional state space model as

$$\boldsymbol{\mathcal{X}}_k = \overline{\boldsymbol{\mathcal{A}}} * \boldsymbol{\mathcal{X}}_{k-1} + \overline{\boldsymbol{\mathcal{B}}} * \boldsymbol{\mathcal{U}}_k \tag{7}$$

$$\boldsymbol{\mathcal{Y}}_k = \boldsymbol{\mathcal{C}} * \boldsymbol{\mathcal{X}}_k + \boldsymbol{\mathcal{D}} * \boldsymbol{\mathcal{U}}_k \tag{8}$$

where $\overline{\boldsymbol{\mathcal{A}}} \in \mathbb{R}^{P \times P \times k_A \times k_A}$ and $\overline{\boldsymbol{\mathcal{B}}} \in \mathbb{R}^{P \times U \times k_B \times k_B}$ denote that these kernels are in discrete-time.

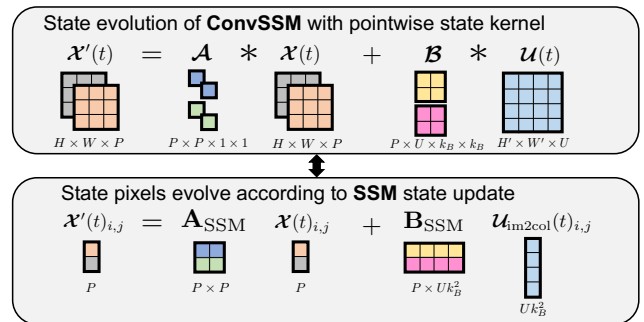

Figure 2: The dynamics of a ConvSSM with pointwise state kernel (top) can be equivalently viewed as the dynamics of an SSM (bottom). See Proposition 3. Each ConvSSM state pixel evolves according to an SSM state update with shared state matrix, $\mathbf{A}_{\mathrm{SSM}}$, and input matrix, $\mathbf{B}_{\mathrm{SSM}}$, that can be formed by reshaping the ConvSSM's state kernel and input kernel. This allows leveraging parameterization insights from deep SSMs [19, 41, 42, 20, 57] to equip ConvS5 to model long-range dependencies.

## 3.2 Parallelizing Convolutional Recurrences

ConvS5 leverages parallel scans to efficiently parallelize the recurrence in (7). As discussed in Section 2.3, this requires a binary associative operator. Given that convolutions are associative, we show:

**Proposition 1.** *Consider a convolutional recurrence as in (7) and define initial parallel scan elements* $c_k = (c_{k,a}, c_{k,b}) := (\overline{\mathcal{A}}, \overline{\mathcal{B}} * \mathcal{U}_k)$. *The binary operator* $\circledast$, *defined below, is associative.*

$$q_i \circledast q_j := (q_{j,a} \circ q_{i,a}, \; q_{j,a} * q_{i,b} \; + \; q_{j,b}), \tag{9}$$

*where* $\circ$ *denotes convolution of two kernels,* $*$ *denotes convolution and* $+$ *is elementwise addition.*

*Proof.* See Appendix A.1. □

Therefore, in theory, we can use this binary operator with a parallel scan to compute the recurrence in (7). However, the binary operator, $\circledast$, requires convolving two $k_A \times k_A$ resolution state kernels together. To maintain equivalence with the sequential scan, the resulting kernel will have resolution $2k_a - 1 \times 2k_a - 1$. This implies that the state kernel will grow during the parallel scan computations for general kernels with a resolution greater than $1 \times 1$. This allows the receptive field to grow in the time direction, a useful feature for capturing spatiotemporal context. However, this kernel growth is computationally infeasible for long sequences.

We address this challenge by taking further inspiration from deep SSMs. These methods opt for simple but computationally advantageous operations in the time direction (linear dynamics) and utilize more complex operations (nonlinear activations) in the depth direction of the model. These nonlinear activations allow a stack of SSM layers with linear dynamics to represent nonlinear systems. Analogously, we choose to use $1 \times 1$ state kernels and perform pointwise state convolutions for the convolutional recurrence of (7). When we stack multiple layers of these ConvSSMs, the receptive field grows in the depth direction of the network and allows the stack of layers to capture the spatiotemporal context [64]. Computationally, we now have a construction that can be parallelized with subquadratic complexity with respect to the sequence length.

**Proposition 2.** *Given the effective inputs* $\overline{\mathcal{B}} * \mathcal{U}_{1:L} \in \mathbb{R}^{L \times H \times W \times P}$ *and a pointwise state kernel* $\mathcal{A} \in \mathbb{R}^{P \times P \times 1 \times 1}$, *the computational cost of computing the convolutional recurrence in Equation 7 with a parallel scan is* $\mathcal{O}\big(L(P^3 + P^2 HW)\big)$.

*Proof.* See Appendix A.2. □

Further, the ConvS5 implementation introduced below admits a diagonalized parameterization that reduces this cost to $\mathcal{O}(LPHW)$. See Section 3.4 and Appendix B for more details.

### 3.3 Connection to State Space Models

Since the convolutions in (5-6) and (7-8) are linear operations, they can be described equivalently as matrix-vector multiplications by flattening the input and state tensors into vectors and using large, circulant matrices consisting of the kernel elements [65]. Thus, any ConvSSM can be described as a large SSM with a circulant dynamics matrix. However, we show here that the use of pointwise state kernels, as described in the previous section, provides an alternative SSM equivalence, which lends a special structure that can leverage the deep SSM parameterization/initialization ideas discussed in Section 2.2 for modeling long-range dependencies. We show that each pixel of the state, $\boldsymbol{\mathcal{X}}(t)_{i,j} \in \mathbb{R}^P$, can be equivalently described as evolving according to a differential equation with a shared state matrix, $\mathbf{A}_{\mathrm{SSM}}$, and input matrix, $\mathbf{B}_{\mathrm{SSM}}$. See Figure 2.

**Proposition 3.** *Consider a ConvSSM state update as in (5) with pointwise state kernel* $\boldsymbol{\mathcal{A}} \in \mathbb{R}^{P \times P \times 1 \times 1}$, *input kernel* $\boldsymbol{\mathcal{B}} \in \mathbb{R}^{P \times U \times k_B \times k_B}$, *and input* $\boldsymbol{\mathcal{U}}(t) \in \mathbb{R}^{H' \times W' \times U}$. *Let* $\boldsymbol{\mathcal{U}}_{\mathrm{im2col}}(t) \in \mathbb{R}^{H \times W \times U k_B^2}$ *be the reshaped result of applying the Image to Column (im2col) [66, 67] operation on the input* $\boldsymbol{\mathcal{U}}(t)$. *Then the dynamics of each state pixel of (5),* $\boldsymbol{\mathcal{X}}(t)_{i,j} \in \mathbb{R}^P$, *evolve according to the following differential equation*

$$\boldsymbol{\mathcal{X}}'(t)_{i,j} = \mathbf{A}_{\mathrm{SSM}}\boldsymbol{\mathcal{X}}(t)_{i,j} + \mathbf{B}_{\mathrm{SSM}}\boldsymbol{\mathcal{U}}_{im2col}(t)_{i,j} \tag{10}$$

*where the state matrix,* $\mathbf{A}_{\mathrm{SSM}} \in \mathbb{R}^{P \times P}$, *and input matrix,* $\mathbf{B}_{\mathrm{SSM}} \in \mathbb{R}^{P \times (U k_B^2)}$, *can be formed by reshaping the state kernel,* $\boldsymbol{\mathcal{A}}$, *and input kernel,* $\boldsymbol{\mathcal{B}}$, *respectively.*

*Proof.* See Appendix A.3. □

Thus, to endow these SSMs with the same favorable long-range dynamical properties as S4/S5 methods, we initialize $\mathbf{A}_{\mathrm{SSM}}$ with a HiPPO [62] inspired matrix and discretize with a learnable timescale parameter to obtain $\overline{\mathbf{A}}_{\mathrm{SSM}}$ and $\overline{\mathbf{B}}_{\mathrm{SSM}}$. Due to the equivalence of Proposition 3, we then reshape these matrices into the discrete ConvSSM state and input kernels of (7) to give the convolutional recurrence the same advantageous dynamical properties. We note that if the input, output and dynamics kernel widths are set to $1 \times 1$, then the ConvSSM formulation is equivalent to "convolving" an SSM across each individual sequence of pixels in the spatiotemporal sequence (this also has connections to the temporal component of S4ND [45] when applied to videos). However, inspired by ConvRNNs, we observed improved performance when leveraging the more general convolutional structure the ConvSSM allows and increasing the input/output kernel sizes to allow local spatial features to be mixed in the dynamical system. See ablations discussed in Section 5.3.

### 3.4 Efficient ConvSSM for Long-Range Dependencies: ConvS5

Here, we introduce *ConvS5*, which combines ideas of parallelization of convolutional recurrences (Section 3.2) and the SSM connection (Section 3.3). ConvS5 is a ConvSSM that leverages parallel scans and deep SSM parameterization/initialization schemes. Given Proposition 3, we implicitly parameterize a pointwise state kernel, $\boldsymbol{\mathcal{A}} \in \mathbb{R}^{P \times P \times 1 \times 1}$ and input kernel $\boldsymbol{\mathcal{B}} \in \mathbb{R}^{P \times U \times k_B \times k_B}$ in (5) using SSM parameters as used by S5 [20], $\mathbf{A}_{\mathrm{S5}} \in \mathbb{R}^{P \times P}$ and $\mathbf{B}_{\mathrm{S5}} \in \mathbb{R}^{P \times (U k_B^2)}$. We discretize these S5 SSM parameters as discussed in Section 2.2 to give

$$\overline{\mathbf{A}}_{\mathrm{S5}} = \mathrm{DISCRETIZE}_{\mathrm{A}}(\mathbf{A}_{\mathrm{S5}}, \boldsymbol{\Delta}), \qquad \overline{\mathbf{B}}_{\mathrm{S5}} = \mathrm{DISCRETIZE}_{\mathrm{B}}(\mathbf{A}_{\mathrm{S5}}, \mathbf{B}_{\mathrm{S5}}, \boldsymbol{\Delta}), \tag{11}$$

and then reshape to give the ConvS5 state update kernels:

$$\overline{\mathbf{A}}_{\mathrm{S5}} \in \mathbb{R}^{P \times P} \xrightarrow{\mathrm{reshape}} \overline{\boldsymbol{\mathcal{A}}}_{\mathrm{ConvS5}} \in \mathbb{R}^{P \times P \times 1 \times 1} \tag{12}$$

$$\overline{\mathbf{B}}_{\mathrm{S5}} \in \mathbb{R}^{P \times (U k_B^2)} \xrightarrow{\mathrm{reshape}} \overline{\boldsymbol{\mathcal{B}}}_{\mathrm{ConvS5}} \in \mathbb{R}^{P \times U \times k_B \times k_B}. \tag{13}$$

We then run the discretized ConvSSM system of (7- 8), using parallel scans to compute the recurrence. In practice, this setup allows us to parameterize ConvS5 using a diagonalized parameterization [41, 42, 20] which reduces the cost of applying the parallel scan in Proposition 2 to $\mathcal{O}(LPHW)$. See Appendix B for a more detailed discussion of parameterization, initialization and discretization.

We define a ConvS5 layer as the combination of ConvS5 with a nonlinear function applied to the ConvS5 outputs. For example, for the experiments in this paper, we use ResNet[68] blocks for the nonlinear activations between layers. However, this is modular, and other choices such as ConvNext [69] or S4ND [45] blocks could easily be used as well. Finally, many ConvS5 layers can be stacked to form a deep spatiotemporal sequence model.

Table 1: Computational complexity of Transformers, ConvRNNS and ConvS5 with respect to sequence length. Metrics are inference cost (cost per step of autoregressive generation), cost per training step, and parallelization ability. ConvS5 combines the best of Transformers and ConvRNNs.

|  | Transformer | ConvRNNs | ConvS5 |
|---|---|---|---|
| Inference | $\mathcal{O}(L)$ | $\mathcal{O}(1)$ | $\mathcal{O}(1)$ |
| Training | $\mathcal{O}(L^2)$ | $\mathcal{O}(L)$ | $\mathcal{O}(L)$ |
| Parallelizable | **Yes** | No | **Yes** |

### 3.5 ConvS5 Properties

Refer to Table 1 for a comparison of computational complexity for Transformers, ConvRNNs and ConvS5. The parallel scans allow ConvS5 to be parallelized across the sequence like a Transformer but with cost only scaling linearly with the sequence length. In addition, ConvS5 is a stateful model like ConvRNNs, allowing fast autoregressive generation, extrapolation to different sequence lengths, and an unbounded context window. The connection with SSMs such as S5 derived in Section 3.3 allows for precisely specifying the initial dynamics to enable the modeling of long-range dependencies and to realize benefits from the unbounded context. Finally, the parallel scan of ConvS5 can allow leveraging the continuous-time parameterization to process irregularly sampled sequences in parallel. S5 achieves this by providing suitable spacing to the discretization operation [20], a procedure that ConvS5 can also use.

We have proposed a general ConvSSM structure that can be easily adapted to future innovations in the deep SSM literature. ConvS5's parallel scan could be used to endow ConvS5 with time-varying dynamics such as the input-dependent dynamics shown to be beneficial in the Liquid-S4 [43] work. Multiple works [51, 52, 50, 53] proposed adding multiplicative gating to allow SSM-based methods to overcome some weaknesses on language modeling. Similar ideas could be useful to ConvSSMs for reasoning over long spatiotemporal sequences.

## 4 Related Work

This work is most closely related to the ConvRNNs and deep SSMs already discussed. We note here that ConvRNNs have been used in numerous domains, including biomedical, robotics, traffic modeling and weather forecasting [70, 1, 71, 2, 3, 72–75, 4, 76, 77]. In addition, many variants have been proposed [78–84, 64]. SSMs have been considered previously for video classification [46, 45, 85], however none addressed the challenging problem of generating long spatiotemporal sequences. S4ND [45] proposed applying separate S4 layers to each axis of an image or video, similar to a PixelRNN [86] approach, and then combining the results with an outer product. While that work mostly focused on images, they also show results for a short 30-frame video classification task. We note this approach could be complementary to ours and the S42D blocks could potentially be used to replace some of the ResNet blocks used as activations in our model. Other model architectures explored in the literature for spatiotemporal modeling include 3D convolution approaches [87–89], transformers, [9, 13, 14, 24–27] and standard RNNs [16, 90]. Attempts to address the problem of modeling long spatiotemporal sequences have involved compressed representations [9, 91, 10, 92, 27, 29], training on sparse subsets of frames [15, 93, 8, 94, 95], temporal hierarchies [16], continuous-time neural representations [93, 95], training on different length sequences at different resolutions [96] and strided sampling [14, 97]. Of particular interest, Yan et al. [13] introduced the Temporally Consistent (TECO) Video Transformer, which achieved state-of-the-art performance on challenging 3D environment benchmarks designed to contain long-range dependencies [13]. This approach combines vector-quantized (VQ) latent dynamics with a MaskGit [98] dynamics prior, and several training tricks to model long videos.

## 5 Experiments

In Section 5.1, we present a long-horizon Moving-MNIST experiment to compare ConvRNNs, Transformers and ConvS5 directly. In Section 5.2, we evaluate ConvS5 on the challenging 3D

Table 2: Quantitative evaluation on the Moving-MNIST dataset [54]. **Top**: To evaluate, we condition on 100 frames, and then show results after generating 800 and 1200 frames. An expanded Table 6 is included in Appendix C with more results, error bars and ablations. Bold scores indicate the best performance and underlined scores indicate the second best performance. **Bottom**: Computational cost comparison for the 600 frame task. Compare to Table 1.

**Trained on 300 frames**

| Method | $100 \rightarrow 800$ | | | | $100 \rightarrow 1200$ | | | |
| --- | --- | --- | --- | --- | --- | --- | --- | --- |
| | FVD ↓ | PSNR ↑ | SSIM ↑ | LPIPS ↓ | FVD ↓ | PSNR ↑ | SSIM ↑ | LPIPS ↓ |
| Transformer [24] | 159 | 12.6 | 0.609 | 0.287 | 265 | 12.4 | 0.591 | 0.321 |
| Performer [33] | 234 | 13.4 | 0.652 | 0.379 | 275 | 13.2 | 0.592 | 0.393 |
| CW-VAE [16] | 104 | 12.4 | 0.592 | 0.277 | **117** | 12.3 | 0.585 | 0.286 |
| ConvLSTM [17] | 128 | 15.0 | 0.737 | 0.169 | 187 | 14.1 | **0.706** | **0.203** |
| ConvS5 | **72** | **16.0** | **0.761** | **0.156** | 187 | **14.5** | 0.678 | 0.230 |

**Trained on 600 frames**

| Method | FVD ↓ | PSNR ↑ | SSIM ↑ | LPIPS ↓ | FVD ↓ | PSNR ↑ | SSIM ↑ | LPIPS ↓ |
| --- | --- | --- | --- | --- | --- | --- | --- | --- |
| Transformer | **42** | 13.7 | 0.672 | 0.207 | 91 | 13.1 | 0.631 | 0.252 |
| Performer | 93 | 12.4 | 0.616 | 0.274 | 243 | 12.2 | 0.608 | 0.312 |
| CW-VAE | 94 | 12.5 | 0.598 | 0.269 | 107 | 12.3 | 0.590 | 0.280 |
| ConvLSTM | 91 | 15.5 | 0.757 | 0.149 | 137 | 14.6 | 0.727 | 0.180 |
| ConvS5 | 47 | **16.4** | **0.788** | **0.134** | **71** | **15.6** | **0.763** | **0.162** |

| | GFLOPS ↓ | Train Step Time (s) ↓ | Train Cost (V100 days) ↓ | Sample Throughput (frames/s) ↑ |
| --- | --- | --- | --- | --- |
| Transformer | 70 | **0.77**(**1.0**×) | **50** | 0.21 (1.0x) |
| ConvLSTM | **65** | 3.0(3.9×) | 150 | **117** (**557**×) |
| ConvS5 | 97 | 0.93(1.2×) | **50** | 90 (429×) |

environment benchmarks proposed in Yan et al. [13]. Finally, in Section 5.3, we discuss ablations that highlight the importance of ConvS5's parameterization.

## 5.1 Long Horizon Moving-MNIST Generation

There are few existing benchmarks for training on and generating long spatiotemporal sequences. We develop a long-horizon Moving-MNIST [54] prediction task that requires training on 300-600 frames and accurately generating up to 1200 frames. This allows for a direct comparison of ConvS5, ConvRNNs and Transformers as well as an efficient attention alternative (Performer [33]) and CW-VAE [16], a temporally hierarchical RNN based method. We first train all models on 300 frames and then evaluate by conditioning on 100 frames before generating 1200. We repeat the evaluation after training on 600 frames. See Appendix D for more experiment details. We present the results after generating 800 and 1200 frames in Table 2. See Appendix C for randomly selected sample trajectories. ConvS5 achieves the best overall performance. When only trained on 300 frames, ConvLSTM and ConvS5 perform similarly when generating 1200 frames, and both outperform the Transformer. All methods benefit from training on the longer 600-frame sequence. However, the longer training length allows ConvS5 to significantly outperform the other methods across the metrics when generating 1200 frames.

In Table 2-bottom we revisit the theoretical properties of Table 1 and compare the empirical computational costs of the Transformer, ConvLSTM and ConvS5 on the 600 frame Moving-MNIST task. Although this specific ConvS5 configuration requires a few more FLOPs due to the convolution computations, ConvS5 is parallelizable during training (unlike ConvLSTM) and has fast autoregressive generation (unlike Transformer) — training 3x faster than ConvLSTM and generating samples 400x faster than Transformers.

## 5.2 Long-range 3D Environment Benchmarks

Yan et al. [13] introduced a challenging video prediction benchmark specifically designed to contain long-range dependencies. This is one of the first comprehensive benchmarks for long-range spatiotemporal modeling and consists of 300 frame videos of agents randomly traversing 3D environ-

Table 3: Quantitative evaluation on the DMLab long-range benchmark [13]. Results from Yan et al. [13] are indicated with ∗. Methods trained using the TECO [13] training framework are at the bottom of the table. TECO methods are slower to sample due to the MaskGit [98] procedure. The expanded Table 8 in Appendix C includes error bars and ablations.

| Method | DMLab | | | | |
| | FVD ↓ | PSNR ↑ | SSIM ↑ | LPIPS ↓ | Sample Throughput (frames/s) ↑ |
| --- | --- | --- | --- | --- | --- |
| FitVid* [90] | 176 | 12.0 | 0.356 | 0.491 | - |
| CW-VAE* [16] | 125 | 12.6 | 0.372 | 0.465 | - |
| Perceiver AR* [39] | 96 | 11.2 | 0.304 | 0.487 | - |
| Latent FDM* [15] | 181 | 17.8 | 0.588 | 0.222 | - |
| Transformer [24] | 97 | _19.9_ | 0.619 | _0.123_ | 9 (1.0×) |
| Performer [33] | _80_ | 17.3 | 0.513 | 0.205 | 7 (0.8×) |
| S5 [20] | 221 | 19.3 | _0.641_ | 0.162 | _28 (3.1×)_ |
| ConvS5 | **66** | **23.2** | **0.769** | **0.079** | **56 (6.2×)** |
| TECO-Transformer* [13] | **28** | _22.4_ | _0.709_ | 0.155 | 16 (1.8×) |
| TECO-Transformer (our run) | **28** | 21.6 | 0.696 | **0.082** | 16 (1.8×) |
| TECO-S5 | 35 | 20.1 | 0.687 | 0.143 | **21 (2.3×)** |
| TECO-ConvS5 | _31_ | **23.8** | **0.803** | _0.085_ | _18 (2.0×)_ |

ments in DMLab [99], Minecraft [100], and Habitat [101] environments. See Appendix C for more experimental details and Appendix E for more details on each dataset.

We train models using the same $16 \times 16$ vector-quantized (VQ) codes from the pretrained VQ-GANs [30] used for TECO and the other baselines in Yan et al. [13]. In addition to ConvS5 and the existing baselines, we also train a Transformer (without the TECO framework), Performer and S5. The S5 baseline serves as an ablation on ConvS5's convolutional tensor-structured approach. Finally, since TECO is essentially a training framework (specialized for Transformers), we also use ConvS5 and S5 layers as a drop-in replacement for the Transformer in TECO. Therefore, we refer to the original version of TECO as *TECO-Transformer*, the ConvS5 version as *TECO-ConvS5* and the S5 version as *TECO-S5*. See Appendix D for detailed information on training procedures, architectures, and hyperparameters.

**DMLab** The results for DMLab are presented in Table 3. Of the methods trained without the TECO framework in the top section of Table 3, ConvS5 outperforms all baselines, including RNN [90, 16], efficient attention [39, 33] and diffusion [15] approaches. ConvS5 also has much faster autoregressive generation than the Transformer. ConvS5 significantly outperforms S5 on all metrics, pointing to the value of the convolutional structure of ConvS5.

For the models trained with the TECO framework, we see that TECO-ConvS5 achieves essentially the same FVD and LPIPS as TECO-Transformer, while significantly improving PSNR and SSIM. Note the sample speed comparisons are less dramatic in this setting since the MaskGit [98] sampling procedure is relatively slow. Still, the sample throughput of TECO-ConvS5 and TECO-S5 remains constant, while TECO-Transformer's throughput decreases with sequence length.

**Minecraft and Habitat** Table 4 presents the results on the Minecraft and Habitat benchmarks. On Minecraft, TECO-ConvS5 achieves the best FVD and performs comparably to TECO-Transformer on the other metrics, outperfoTarming all other baselines. On Habitat, TECO-ConvS5 is the only method to achieve a comparable FVD to TECO-Transformer, while outperforming it on PSNR and SSIM.

## 5.3 ConvS5 ablations

In Table 5 we present ablations on the convolutional structure of ConvS5. We compare different input and output kernel sizes for the ConvSSM and also compare the default ResNet activations to a channel mixing GLU [102] activation. Where possible, when reducing the sizes of the ConvSSM kernels, we redistribute parameters to the ResNet kernels or the GLU sizes to compare similar parameter counts. The results suggest more convolutional structure improves performance.

Table 4: Quantitative evaluation on the Minecraft and Habitat long-range benchmarks [13]. Results from Yan et al. [13] are indicated with ∗. See expanded Table 10 with error bars in Appendix C.

| Method | Minecraft | | | | Habitat | | | |
| | FVD ↓ | PSNR ↑ | SSIM ↑ | LPIPS ↓ | FVD ↓ | PSNR ↑ | SSIM ↑ | LPIPS ↓ |
|---|---|---|---|---|---|---|---|---|
| FitVid* | 956 | 13.0 | 0.343 | 0.519 | - | - | - | - |
| CW-VAE* | 397 | 13.4 | 0.338 | 0.441 | - | - | - | - |
| Perceiver AR* | 76 | 13.2 | 0.323 | 0.441 | 164 | 12.8 | **0.405** | 0.676 |
| Latent FDM* | 167 | 13.4 | 0.349 | 0.429 | 433 | 12.5 | 0.311 | **0.582** |
| TECO-Transformer* | 116 | **15.4** | **0.381** | **0.340** | **76** | 12.8 | 0.363 | 0.604 |
| TECO-ConvS5 | **71** | 14.8 | 0.374 | 0.355 | 95 | **12.9** | 0.390 | 0.632 |

Table 5: Ablations of ConvS5 convolutional structure for DMLab long-range benchmark dataset [13]. More convolutional structure improves overall performance. See expanded Table 9 in Appendix.

| conv. | $\mathcal{B}$ kernel | $\mathcal{C}$ kernel | DMLab nonlinearity | FVD ↓ | PSNR ↑ | SSIM ↑ | LPIPS ↓ |
|---|---|---|---|---|---|---|---|
| x | - | - | GLU | 221 | 19.3 | 0.641 | 0.162 |
| o | $1 \times 1$ | $1 \times 1$ | GLU | 187 | 21.0 | 0.689 | 0.112 |
| o | $1 \times 1$ | $5 \times 5$ | GLU | 89 | 21.5 | 0.713 | 0.106 |
| o | $3 \times 3$ | $3 \times 3$ | GLU | 96 | 22.7 | 0.762 | 0.088 |
| o | $1 \times 1$ | $1 \times 1$ | ResNet | 81 | 23.0 | 0.767 | 0.083 |
| o | $1 \times 1$ | $3 \times 3$ | ResNet | 68 | 22.8 | 0.756 | 0.085 |
| o | $3 \times 3$ | $3 \times 3$ | ResNet | **67** | **23.2** | **0.769** | **0.079** |

We also perform ablations to evaluate the importance of ConvS5's deep SSM-inspired parameterization/initialization. We evaluate the performance of a ConvSSM with randomly initialized state kernel on both Moving-Mnist and DMLab. See Table 6 and Table 8 in Appendix C. We observe a degradation in performance in all settings for this ablation. This reflects prior results for deep SSMs [56, 19, 20, 57] and highlights the importance of the connection developed in Section 3.3.

# 6 Discussion

This work introduces ConvS5, a new spatiotemporal modeling layer that combines the fast, stateful autoregressive generation of ConvRNNs with the ability to be parallelized across the sequence like Transformers. Its computational cost scales linearly with the sequence length, providing better scaling for longer spatiotemporal sequences. ConvS5 also leverages insights from deep SSMs to model long-range dependencies effectively.

We note that despite the ConvS5's sub-quadratic scaling, it did not show significant training speedups over Transformers for sequence lengths of 300-600 frames. (See detailed run-time comparisons in Appendix C.) We expect ConvS5 to excel in training efficiency when applied to much longer spatiotemporal sequences where the quadratic scaling of Transformers dominates. We hope this work inspires the creation of longer spatiotemporal datasets and benchmarks. At the current sequence lengths, future optimizations of the parallel scan implementations in common deep learning frameworks will be helpful. In addition, the ResNet blocks used as the activations between ConvS5 layers could be replaced with efficient activations such as sparse convolutions [103] or S4ND [45].

An interesting future direction is to further utilize ConvS5's continuous-time parameterization. Deep SSMs show strong performance when trained at one resolution and evaluated on another [19, 41, 42, 20, 43, 45]. In addition, S5 can leverage its parallel scan to effectively model irregularly sampled sequences [20]. ConvS5 can be used for such applications in the spatiotemporal domain [104, 93, 95]. Finally, ConvS5 is modular and flexible. We have demonstrated that ConvS5 works well as a drop-in replacement in the TECO [13] training framework specifically developed for Transformers. Due to its favorable properties, we expect ConvS5 to also serve as a building block of new approaches for modeling much longer spatiotemporal sequences and multimodal applications.

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

# Appendix for: Convolutional State Space Models for Long-range Spatiotemporal Modeling

**Contents:**

- **Appendix A**: Propositions
- **Appendix B**: ConvS5 Details: Parameterization, Initialization, Discretization
- **Appendix C**: Supplementary Results
- **Appendix D**: Experiment Configurations
- **Appendix E**: Datasets

# A  Propositions

## A.1  Parallel Scan for Convolutional Recurrences

**Proposition 1.** *Consider a convolutional recurrence as in (7) and define initial parallel scan elements* $c_k = (c_{k,a}, c_{k,b}) := (\overline{\mathcal{A}}, \overline{\mathcal{B}} * \mathcal{U}_k)$. *The binary operator* $\circledast$, *defined below, is associative.*

$$c_i \circledast c_j := (c_{j,a} \circ c_{i,a}, \; c_{j,a} * c_{i,b} \; + \; c_{j,b}), \tag{14}$$

*where* $\circ$ *denotes convolution of two kernels,* $*$ *denotes convolution between a kernel and input, and* $+$ *is elementwise addition.*

*Proof.* Using that $\circ$ is associative and the companion operator of $*$, i.e. $(d \circ e) * f = d * (e * f)$ (see Blelloch [63], Section 1.4), we have:

$$(c_i \circledast c_j) \circledast c_k = (c_{j,a} \circ c_{i,a}, \; c_{j,a} * c_{i,b} \; + \; c_{j,b}) \circledast (c_{k,a}, c_{k,b}) \tag{15}$$

$$= \left( c_{k,a} \circ (c_{j,a} \circ c_{i,a}), \; c_{k,a} * (c_{j,a} * c_{i,b} + c_{j,b}) + c_{k,b} \right) \tag{16}$$

$$= \left( (c_{k,a} \circ c_{j,a}) \circ c_{i,a}, \; c_{k,a} * (c_{j,a} * c_{i,b}) + c_{k,a} * c_{j,b} + c_{k,b} \right) \tag{17}$$

$$= \left( (c_{k,a} \circ c_{j,a}) \circ c_{i,a}, \; (c_{k,a} \circ c_{j,a}) * c_{i,b} + c_{k,a} * c_{j,b} + c_{k,b} \right) \tag{18}$$

$$= c_i \circledast (c_{k,a} \circ c_{j,a}, \; c_{k,a} * c_{j,b} + c_{k,b}) \tag{19}$$

$$= c_i \circledast (c_j \circledast c_k) \tag{20}$$

$\square$

## A.2  Computational Cost of Parallel Scan for Convolutional Recurrences

**Proposition 2.** *Given the effective inputs* $\overline{\mathcal{B}} * \mathcal{U}_{1:L} \in \mathbb{R}^{L \times H \times W \times P}$ *and a pointwise state kernel* $\overline{\mathcal{A}} \in \mathbb{R}^{P \times P \times 1 \times 1}$, *the computational cost of computing the convolutional recurrence in Equation 7 with a parallel scan is* $\mathcal{O}\big(L(P^3 + P^2 HW)\big)$.

*Proof.* Following Blelloch [63], given a single processor, the cost of computing the recurrence sequentially using the binary operator $\circledast$ defined in Proposition 1 is $\mathcal{O}\big(L(T_\circ + T_* + T_+)\big)$ where $T_\circ$ refers to the cost of convolving two kernels, $T_*$ is the cost of convolution between a kernel and input and $T_+$ is the cost of elementwise addition. The cost of elementwise addition is $T_+ = \mathcal{O}(PHW)$. For state kernels with resolution $k_A$, $T_\circ = \mathcal{O}(P^3 k_A^4)$ and $T_* = \mathcal{O}(P^2 k_A^2 HW)$. For pointwise convolutions this becomes $T_\circ = \mathcal{O}(P^3)$ and $T_* = \mathcal{O}(P^2 HW)$. Thus, the cost of computing the recurrence sequentially using $\circledast$ is $\mathcal{O}\big(L(P^3 + P^2 HW)\big)$. Since there are work-efficient algorithms for parallel scans [105], the overall cost of the parallel scan is also $\mathcal{O}\big(L(P^3 + P^2 HW)\big)$. $\square$

Note that ConvS5's diagonalized parameterization discussed in Section 3.4 and Appendix B leads to $T_\circ = \mathcal{O}(P)$ and $T_* = \mathcal{O}(PHW)$. Therefore the cost of applying the parallel scan with ConvS5 is $\mathcal{O}(LPHW)$.

## A.3  Connection Between ConvSSMs and SSMs

**Proposition 3.** *Consider a ConvSSM state update as in (5) with pointwise state kernel* $\mathcal{A} \in \mathbb{R}^{P \times P \times 1 \times 1}$, *input kernel* $\mathcal{B} \in \mathbb{R}^{P \times U \times k_B \times k_B}$, *and input* $\mathcal{U}(t) \in \mathbb{R}^{H' \times W' \times U}$. *Let* $\mathcal{U}_{\mathrm{im2col}}(t) \in \mathbb{R}^{H \times W \times U k_B^2}$ *be the reshaped result of applying the Image to Column (im2col) [66, 67] operation on the input* $\mathcal{U}(t)$. *Then the dynamics of each state pixel of (5),* $\mathcal{X}(t)_{i,j} \in \mathbb{R}^P$, *evolve according to the following differential equation*

$$\mathcal{X}'(t)_{i,j} = \mathbf{A}_{\mathrm{SSM}} \mathcal{X}(t)_{i,j} + \mathbf{B}_{\mathrm{SSM}} \mathcal{U}_{im2col}(t)_{i,j} \tag{21}$$

*where the state matrix,* $\mathbf{A}_{\mathrm{SSM}} \in \mathbb{R}^{P \times P}$, *and input matrix,* $\mathbf{B}_{\mathrm{SSM}} \in \mathbb{R}^{P \times (U k_B^2)}$, *can be formed by reshaping the state kernel,* $\mathcal{A}$, *and input kernel,* $\mathcal{B}$, *respectively.*

*Proof.* Let $\mathbf{U}_{\text{im2col}} \in \mathbb{R}^{Uk_b^2 \times HW}$ denote the result of performing the im2col operation on the input $\mathcal{U}(t)$ for convolution with the kernel $\mathcal{B}$. Reshape this matrix into the tensor $\mathcal{U}_{\text{im2col}}(t) \in \mathbb{R}^{H \times W \times Uk_b^2}$. Reshape $\mathcal{U}_{\text{im2col}}(t)$ once more into the tensor $\mathcal{V}(t) \in \mathbb{R}^{H \times W \times U \times k_B \times k_B}$.

Now, we can write the evolution for the individual channels of each pixel, $\mathcal{X}'(t)_{i,j,k}$, in (5) as

$$\mathcal{X}'(t)_{i,j,k} = \sum_{l=0}^{P-1} \mathcal{A}_{k,l,0,0} \mathcal{X}(t)_{i,j,l} + \sum_{q=0}^{U-1} \sum_{m=0}^{k_B-1} \sum_{n=0}^{k_B-1} \mathcal{B}_{k,q,m,n} \mathcal{V}(t)_{i,j,q,m,n}. \tag{22}$$

Let $\mathbf{A}_{\text{SSM}} \in \mathbb{R}^{P \times P}$ be a matrix with rows, $\mathbf{A}_{\text{SSM},i} \in \mathbb{R}^P$, corresponding to a flattened version of the output features of $\mathcal{A}$, i.e. $\mathcal{A}_i \in \mathbb{R}^{P \times 1 \times 1}$. Similarly, reshape $\mathcal{B}$ into a matrix $\mathbf{B}_{\text{SSM}} \in \mathbb{R}^{P \times (Uk_B^2)}$ where the rows, $\mathbf{B}_{\text{SSM},i} \in \mathbb{R}^{Uk_B^2}$ correspond to a flattened version of the output features of $\mathcal{B}$, i.e. $\mathcal{B}_i \in \mathbb{R}^{U \times k_B \times k_B}$.

Then we can rewrite (22) equivalently as

$$\mathcal{X}'(t)_{i,j,k} = \sum_{l=0}^{P-1} \mathcal{A}_{k,l,0,0} \mathcal{X}(t)_{i,j,l} + \sum_{q=0}^{U-1} \sum_{m=0}^{k_B-1} \sum_{n=0}^{k_B-1} \mathcal{B}_{k,q,m,n} \mathcal{V}(t)_{i,j,q,m,n} \tag{23}$$

$$= \sum_{l=0}^{P-1} \mathbf{A}_{\text{SSM},k,l} \mathcal{X}(t)_{i,j,l} + \sum_{v=0}^{Uk_B^2-1} \mathbf{B}_{\text{SSM},k,v} \mathcal{U}_{\text{im2col}}(t)_{i,j,v} \tag{24}$$

$$= \mathbf{A}_{\text{SSM},k}^T \mathcal{X}(t)_{i,j} + \mathbf{B}_{\text{SSM},k}^T \mathcal{U}_{\text{im2col}}(t)_{i,j} \tag{25}$$

$\square$

# B    ConvS5 Details: Parameterization, Discretization, Initialization

## B.1    Background: S5

**S5 Parameterization and Discretization**    S5 [20] uses a diagonalized parameterization of the general SSM in (3).

Let $\mathbf{A}_{S5} = \mathbf{V}\mathbf{\Lambda}_{S5}\mathbf{V}^{-1} \in \mathbb{R}^{P \times P}$ where $\mathbf{\Lambda}_{S5} \in \mathbb{C}^{P \times P}$ is a complex-valued diagonal matrix and $\mathbf{V} \in \mathbb{C}^{P \times P}$ corresponds to the eigenvectors. Defining $\tilde{\mathbf{x}}(t) = \mathbf{V}^{-1}\mathbf{x}(t)$, $\tilde{\mathbf{B}} = \mathbf{V}^{-1}\mathbf{B}$, and $\tilde{\mathbf{C}} = \mathbf{CV}$ we can reparameterize the SSM of (3) as the diagonalized system:

$$\frac{\mathrm{d}\tilde{\mathbf{x}}(t)}{\mathrm{d}t} = \mathbf{\Lambda}_{S5}\tilde{\mathbf{x}}(t) + \tilde{\mathbf{B}}\mathbf{u}(t), \qquad \mathbf{y}(t) = \tilde{\mathbf{C}}\tilde{\mathbf{x}}(t) + \mathbf{D}\mathbf{u}(t). \tag{26}$$

S5 uses learnable timescale parameters $\mathbf{\Delta} \in \mathbb{R}^P$ and the following zero-order hold (ZOH) disretization:

$$\overline{\mathbf{\Lambda}}_{S5} = \mathrm{DISCRETIZE_A}(\mathbf{\Lambda}_{S5}, \mathbf{\Delta}) := e^{\mathbf{\Lambda}_{S5}\mathbf{\Delta}} \tag{27}$$

$$\overline{\mathbf{B}}_{S5} = \mathrm{DISCRETIZE_B}(\mathbf{\Lambda}_{S5}, \tilde{\mathbf{B}}, \mathbf{\Delta}) := \mathbf{\Lambda}_{S5}^{-1}(\overline{\mathbf{\Lambda}}_{S5} - \mathbf{I})\tilde{\mathbf{B}} \tag{28}$$

**S5 Initialization**    S5 initializes its state matrix by diagonalizing $\mathbf{A}_{S5}$ as defined here:

$$\mathbf{A}_{S5_{nk}} = - \begin{cases} (n + \frac{1}{2})^{1/2}(k + \frac{1}{2})^{1/2}, & n > k \\ \frac{1}{2}, & n = k \\ (n + \frac{1}{2})^{1/2}(k + \frac{1}{2})^{1/2}, & n < k \end{cases}. \tag{29}$$

This matrix is the normal part of the normal plus low-rank HiPPO-LegS matrix from the HiPPO framework [62] for online function approximation. S4 originally initialized its single-input, single-output (SISO) SSMs with a representation of the full HiPPO-LegS matrix. This was shown to be approximating long-range dependencies at initialization with respect to an infinitely long, exponentially-decaying measure [106]. Gupta et al. [41] empirically showed that the low-rank terms could be removed without impacting performance. Gu et al. [42] showed that in the limit of infinite state dimension, the linear, single-input ODE with this normal approximation to the HiPPO-LegS matrix produces the same dynamics as the linear, single-input ODE with the full HiPPO-LegS matrix. The S5 work extended these findings to the multi-input SSM setting [20].

**Importance of SSM Parameterization, Discretization and Initialization**    Prior research has highlighted the importance of parameterization, discretization and initialization choices of deep SSM methods through ablations and analysis [56, 19, 42, 20, 57]. Concurrent work from Orvieto et al. [57] provides particular insight into the favorable initial eigenvalue distributions provided by initializing with HiPPO-inspired matrices as well as an important normalization effect provided by the explicit discretization procedure. They also introduce a purely discrete-time parameterization that can perform similarly to the continuous-time discretization of S4 and S5. However, their parameterization practically ends up quite similar to the equations of (27-28). We choose to use the continuous-time parameterization of S5 for the implicit parameterization of ConvS5 since it can also be leveraged for zero-shot resolution changes [19, 20, 45] and processing irregularly sampled time-series in parallel [20]. However, due to Proposition 3, other long-range SSM parameterization strategies can also be used, such as in Orvieto et al. [57] or potential future innovations.

## B.2    ConvS5 Diagonalization

We leverage S5's diagonalized parameterization to reduce the cost of the parallel scan of ConvS5.

Concretely, we initialize $\mathbf{A}_{S5}$ as in (29) and diagonalize as $\mathbf{A}_{S5} = \mathbf{V}\mathbf{\Lambda}_{S5}\mathbf{V}^{-1}$. To apply ConvS5, we compute $\overline{\mathbf{\Lambda}}_{S5}$ and $\overline{\mathbf{B}}_{S5}$ using (27-28), and then form the ConvS5 state and input kernels:

$$\overline{\mathbf{\Lambda}}_{S5} \in \mathbb{R}^{P \times P} \xrightarrow{\mathrm{reshape}} \overline{\mathcal{A}}_{\mathrm{ConvS5}} \in \mathbb{R}^{P \times P \times 1 \times 1} \tag{30}$$

$$\overline{\mathbf{B}}_{S5} \in \mathbb{R}^{P \times (Uk_B^2)} \xrightarrow{\mathrm{reshape}} \overline{\mathcal{B}}_{\mathrm{ConvS5}} \in \mathbb{R}^{P \times U \times k_B \times k_B}. \tag{31}$$

See Listing 1 for an example of the core implementation. Note, the state kernel $\overline{\mathcal{A}}_{\mathrm{ConvS5}}$ is "diagonalized" in the sense that all entries in the state kernel are zero except $\overline{\mathcal{A}}_{\mathrm{ConvS5},i,i} = \overline{\mathbf{\Lambda}}_{S5,i,i} \ \forall i \in [P]$.

This means that the pointwise convolutions reduce to channel-wise multiplications. However, this does not reduce expressivity compared to a full pointwise convolution. This is because, given the ConvSSM to SSM equivalence of Proposition 3 and the use of complex-valued kernels, the diagonalization maintains expressivity since almost all SSMs are diagonalizable [41, 42], which follows from the well-known fact that almost all square matrices diagonalize over the complex plane.

```
1    import jax
2    import jax.numpy as np
3    from ConvSSM_helpers import discretize, Conv2D, ResNet_Block
4    parallel_scan = jax.lax.associative_scan
5
6    def apply_ConvS5_layer(A, B, B_shape, C_kernel, log_Delta, resnet_params, us, x0):
7        """Compute the outputs of ConvS5 layer given input sequence.
8        Args:
9            A (complex64): S5 state matrix                   (P,)
10           B (complex64): S5 input matrix                   (P,Uk_B^2)
11           B_shape (tuple): shape of B_kernel
12           C_kernel (complex64) output kernel              (U,P,k_C,k_C)
13           log_Delta (float32): learnable timescale params (P,)
14           resnet_params (dict): ResNet block params
15           us (float32): input sequence of features        (L,bsz,H,W,U)
16           x0 (complex64): initial state                   (bsz,H,W,P)
17       Returns:
18           outputs (float32): the ConvS5 layer outputs     (L,bsz,H,W,U)
19           x_L (complex64): the last state of the ConvSSM  (bsz,H,W,P)
20       """
21       # Discretize and reshape ConvS5 state and input kernels
22       P, U, k_B = B_shape
23       A_bar, B_bar = discretize(A, B, np.exp(log_Delta))
24       A_kernel = A_bar    # already correct shape due to diagonalization
25       B_kernel = B_bar.reshape(P, U, k_B, k_B)
26
27       # Apply ConvS5
28       ys, xs = apply_ConvS5(A_kernel, B_kernel, C_kernel, us, x0)
29
30       # Apply ResNet activation function
31       outputs = jax.vmap(ResNet_Block, axis=(None,0))(resnet_params, ys)
32       return outputs, xs[-1]
33
34   def apply_ConvS5(A_kernel, B_kernel, C_kernel, us, x0):
35       """Compute the output sequence of the convolutional SSM
36          given the input sequence using a parallel scan.
37          Computes x_k = A * x_{k-1} + B * u_k
38                   y_k = C * x_k
39          where * is a convolution operator.
40       Args:
41           A_kernel (complex64): state kernel   (P,)
42           B_kernel (complex64): input kernel   (P,U,k_B,k_B)
43           C_kernel (complex64): output kernel  (U,P,k_C,k_C)
44           us (float32): input sequence         (L,bsz,H,W,U)
45           x0 (complex64): initial state        (bsz,H,W,P)
46       Returns:
47           ys (float32): the convS5 outputs     (L,bsz,H,W,U)
48           x_L (complex64): the last state      (bsz,H,W,P)
49       """
50       # Compute initial scan elements
51       As = np.repeat(A_kernel[None, ...], us.shape[0], axis=0)
52       Bus = jax.vmap(Conv2D)(B_kernel, np.complex64(us))
53       Bus = Bus.at[0].add(np.expand_dims(A_bar, (0, 1, 2)) * x0)
54
55       # Convolutional recurrence with parallel scan
56       _, xs = parallel_scan(conv_binary_operator, (As, Bus))
57
58       # Compute ConvS5 outputs
59       ys = jax.vmap(Conv2D)(C_kernel, xs).real
60       return ys, xs
61
62   def conv_binary_operator(q_i, q_j):
63       """Binary operator for convolutional recurrence
64          with "diagonalized" 1X1 state kernels.
65       Args:
66           q_i, q_j (tuples): scan elements q_i=(A_i, BU_i) where
67                   A_i (complex64) is state kernel      (P,)
68                   BU_i (complex64) is effective input  (bsz,H,W,P)
69       Returns:
70           output tuple q_i \circledast q_j
71       """
72       A_i, BU_i = q_i
73       A_j, BU_j = q_j
74       # Convolve "diagonal" 1X1 kernels
75       AA = A_j * A_i
76       # Convolve "diagonal" A_j with BU_i
77       A_jBU_i = np.expand_dims(A_j, (0, 1, 2)) * BU_i
78       return AA, A_jBU_i + BU_j
```

Listing 1: JAX implementation of core code to apply a single ConvS5 layer to a batch of spatiotemporal input sequences.

# C  Supplementary Results

We include expanded tables and sample trajectories from the experiments in the main paper. Sample videos can be found at:

https://sites.google.com/view/convssm.

## C.1  Moving-MNIST

Table 6: Full results on the Moving-MNIST dataset [54]. For the top table, all models are trained on 300 frames. For the bottom table, all models are trained on 600 frames. The evaluation task is to condition on 100 frames, and then generate forward 400, 800 and 1200 frames. ConvSSM (ablation) is performed by randomly initializing the state kernel (see Section 5.3 and Appendix D.4).

**Trained on 300 frames**

| Method | Params | $100 \to 400$ | | | | $100 \to 800$ | | | | $100 \to 1200$ | | | |
| | | FVD ↓ | PSNR ↑ | SSIM ↑ | LPIPS ↓ | FVD ↓ | PSNR ↑ | SSIM ↑ | LPIPS ↓ | FVD ↓ | PSNR ↑ | SSIM ↑ | LPIPS ↓ |
|---|---|---|---|---|---|---|---|---|---|---|---|---|---|
| Transformer | 164M | $73 \pm 3$ | $13.5 \pm 0.1$ | $0.669 \pm 0.002$ | $0.213 \pm 0.003$ | $159 \pm 7$ | $12.6 \pm 0.1$ | $0.609 \pm 0.002$ | $0.287 \pm 0.001$ | $265 \pm 8$ | $12.4 \pm 0.1$ | $0.591 \pm 0.002$ | $0.321 \pm 0.002$ |
| Performer | 164M | $111 \pm 9$ | $13.4 \pm 0.1$ | $0.653 \pm 0.002$ | $0.288 \pm 0.001$ | $234 \pm 1$ | $13.4 \pm 0.1$ | $0.652 \pm 0.006$ | $0.379 \pm 0.002$ | $275 \pm 5$ | $13.2 \pm 0.1$ | $0.592 \pm 0.001$ | $0.393 \pm 0.001$ |
| CW-VAE | 20M | $78 \pm 1$ | $12.7 \pm 0.1$ | $0.611 \pm 0.002$ | $0.254 \pm 0.001$ | $104 \pm 2$ | $12.4 \pm 0.1$ | $0.592 \pm 0.002$ | $0.277 \pm 0.002$ | $\mathbf{117 \pm 2}$ | $12.3 \pm 0.1$ | $0.585 \pm 0.002$ | $0.286 \pm 0.001$ |
| ConvLSTM | 20M | $57 \pm 3$ | $16.9 \pm 0.2$ | $0.796 \pm 0.004$ | $0.113 \pm 0.002$ | $128 \pm 4$ | $15.0 \pm 0.1$ | $0.737 \pm 0.003$ | $0.169 \pm 0.001$ | $187 \pm 6$ | $14.1 \pm 0.1$ | $\mathbf{0.706 \pm 0.003}$ | $\mathbf{0.203 \pm 0.001}$ |
| ConvSSM (ablation) | 20M | $67 \pm 3$ | $15.5 \pm 0.1$ | $0.742 \pm 0.001$ | $0.168 \pm 0.001$ | $287 \pm 5$ | $13.6 \pm 0.1$ | $0.577 \pm 0.001$ | $0.293 \pm 0.001$ | $511 \pm 8$ | $13.3 \pm 0.1$ | $0.515 \pm 0.001$ | $0.348 \pm 0.001$ |
| ConvS5 | 20M | $\mathbf{26 \pm 1}$ | $\mathbf{18.1 \pm 0.1}$ | $\mathbf{0.830 \pm 0.003}$ | $\mathbf{0.094 \pm 0.002}$ | $\mathbf{72 \pm 3}$ | $\mathbf{16.0 \pm 0.1}$ | $\mathbf{0.761 \pm 0.005}$ | $\mathbf{0.156 \pm 0.003}$ | $\underline{187 \pm 5}$ | $\mathbf{14.5 \pm 0.1}$ | $\underline{0.678 \pm 0.003}$ | $\underline{0.230 \pm 0.004}$ |

**Trained on 600 frames**

| Method | Params | $100 \to 400$ | | | | $100 \to 800$ | | | | $100 \to 1200$ | | | |
| | | FVD ↓ | PSNR ↑ | SSIM ↑ | LPIPS ↓ | FVD ↓ | PSNR ↑ | SSIM ↑ | LPIPS ↓ | FVD ↓ | PSNR ↑ | SSIM ↑ | LPIPS ↓ |
|---|---|---|---|---|---|---|---|---|---|---|---|---|---|
| Transformer | 164M | $\mathbf{21 \pm 1}$ | $15.0 \pm 0.1$ | $0.741 \pm 0.002$ | $0.138 \pm 0.001$ | $\mathbf{42 \pm 2}$ | $13.7 \pm 0.1$ | $0.672 \pm 0.002$ | $0.207 \pm 0.003$ | $\underline{91 \pm 6}$ | $13.1 \pm 0.1$ | $0.631 \pm 0.004$ | $0.252 \pm 0.002$ |
| Performer | 164M | $27 \pm 1$ | $13.1 \pm 0.1$ | $0.654 \pm 0.004$ | $0.206 \pm 0.001$ | $93 \pm 5$ | $12.4 \pm 0.1$ | $0.616 \pm 0.002$ | $0.274 \pm 0.001$ | $243 \pm 7$ | $12.2 \pm 0.1$ | $0.608 \pm 0.001$ | $0.312 \pm 0.002$ |
| CW-VAE | 20M | $73 \pm 2$ | $12.9 \pm 0.1$ | $0.621 \pm 0.004$ | $0.242 \pm 0.001$ | $94 \pm 3$ | $12.5 \pm 0.9$ | $0.598 \pm 0.004$ | $0.269 \pm 0.001$ | $107 \pm 2$ | $12.3 \pm 0.1$ | $0.590 \pm 0.004$ | $0.280 \pm 0.002$ |
| ConvLSTM | 20M | $39 \pm 5$ | $17.3 \pm 0.2$ | $0.812 \pm 0.005$ | $0.100 \pm 0.003$ | $91 \pm 7$ | $15.5 \pm 0.2$ | $0.757 \pm 0.005$ | $0.149 \pm 0.003$ | $137 \pm 9$ | $14.6 \pm 0.1$ | $0.727 \pm 0.004$ | $0.180 \pm 0.003$ |
| ConvSSM (ablation) | 20M | $81 \pm 6$ | $15.5 \pm 0.1$ | $0.743 \pm 0.002$ | $0.163 \pm 0.003$ | $145 \pm 8$ | $14.3 \pm 0.1$ | $0.696 \pm 0.002$ | $0.218 \pm 0.002$ | $215 \pm 9$ | $13.4 \pm 0.1$ | $0.614 \pm 0.001$ | $0.287 \pm 0.001$ |
| ConvS5 | 20M | $\underline{23 \pm 3}$ | $\mathbf{18.1 \pm 0.1}$ | $\mathbf{0.832 \pm 0.003}$ | $\mathbf{0.092 \pm 0.003}$ | $\underline{47 \pm 7}$ | $\mathbf{16.4 \pm 0.1}$ | $\mathbf{0.788 \pm 0.002}$ | $\mathbf{0.134 \pm 0.003}$ | $\mathbf{71 \pm 9}$ | $\mathbf{15.6 \pm 0.1}$ | $\mathbf{0.763 \pm 0.002}$ | $\mathbf{0.162 \pm 0.003}$ |

Table 7: Model runtime comparison for Moving-MNIST results in Table 6. ConvS5 can be parallelized like a Transformer but maintains the constant cost-per-step autoregressive generation of ConvRNNs.

| Method | Parallelizable | $100 \to 400$ Train Step Time (s) ↓ | $100 \to 400$ Sample Throughput (frames/s) ↑ | $100 \to 800$ Sample Throughput (frames/s) ↑ | $100 \to 1200$ Sample Throughput (frames/s) ↑ |
|---|---|---|---|---|---|
| Transformer | **YES** | $\mathbf{0.77}$ ($\mathbf{1.0\times}$) | $1.1$ ($1.0\times$) | $0.34$ ($1.0\times$) | $0.21$ ($1.0\times$) |
| ConvLSTM | NO | $3.0$ ($3.9\times$) | $\mathbf{117}$ ($\mathbf{106\times}$) | $\mathbf{117}$ ($\mathbf{345\times}$) | $\mathbf{117}$ ($\mathbf{557\times}$) |
| ConvS5 | **YES** | $\underline{0.93}$ ($1.2\times$) | $\underline{90}$ ($82\times$) | $90$ ($265\times$) | $\underline{90}$ ($429\times$) |

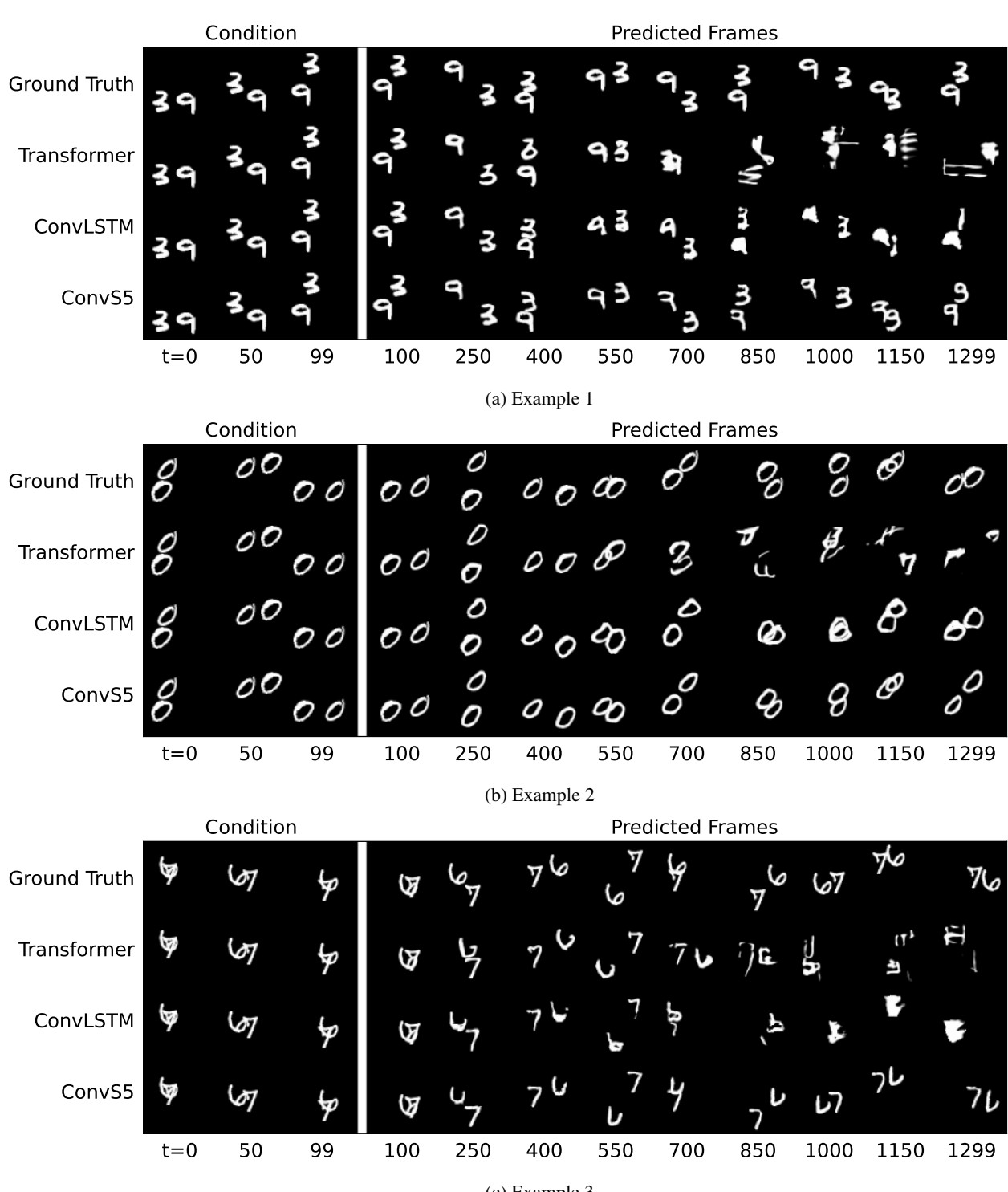

(a) Example 1

(b) Example 2

(c) Example 3

Figure 3: Moving-MNIST Samples: 1200 frames generated conditioned on 100.

## C.2 3D Environments

Table 8: Full results for DMLab long-range benchmark dataset [13]. Results from Yan et al. [13] are indicated with ∗. We separate out the methods trained using the TECO [13] training framework in the bottom of the table. TECO-ConvSSM (ablation) refers to the ablation performed by randomly initializing the state kernel (see Section 5.3 and Appendix D.5).

| Method | Params | DMLab | | | |
| --- | --- | --- | --- | --- | --- |
| | | FVD ↓ | PSNR ↑ | SSIM ↑ | LPIPS ↓ |
| FitVid* | 165M | $176 \pm 4.86$ | $12.0 \pm 0.013$ | $0.356 \pm 0.00171$ | $0.491 \pm 0.00108$ |
| CW-VAE* | 111M | $125 \pm 7.95$ | $12.6 \pm 0.059$ | $0.372 \pm 0.00033$ | $0.465 \pm 0.00156$ |
| Perceiver AR* | 30M | $96.3 \pm 3.64$ | $11.2 \pm 0.004$ | $0.304 \pm 0.00004$ | $0.487 \pm 0.00123$ |
| Latent FDM* | 31M | $181 \pm 2.20$ | $17.8 \pm 0.111$ | $0.588 \pm 0.00453$ | $0.222 \pm 0.00493$ |
| Transformer | 152M | $97.0 \pm 5.98$ | $\underline{19.9 \pm 0.108}$ | $0.619 \pm 0.00506$ | $\underline{0.123 \pm 0.00191}$ |
| Performer | 152M | $\underline{80.3 \pm 3.21}$ | $17.3 \pm 0.074$ | $0.513 \pm 0.00492$ | $0.205 \pm 0.00315$ |
| S5 | 140M | $221 \pm 13.1$ | $19.3 \pm 0.128$ | $\underline{0.641 \pm 0.00400}$ | $0.162 \pm 0.04510$ |
| ConvS5 | 100M | $\mathbf{66.6 \pm 4.81}$ | $\mathbf{23.2 \pm 0.053}$ | $\mathbf{0.769 \pm 0.01020}$ | $\mathbf{0.079 \pm 0.00073}$ |
| TECO-Transformer* | 173M | $\mathbf{27.5 \pm 1.77}$ | $\underline{22.4 \pm 0.368}$ | $\underline{0.709 \pm 0.0119}$ | $0.155 \pm 0.00958$ |
| TECO-Transformer (our run) | 173M | $\mathbf{28.2 \pm 0.66}$ | $21.6 \pm 0.079$ | $0.696 \pm 0.02640$ | $\mathbf{0.082 \pm 0.00119}$ |
| TECO-S5 | 180M | $34.6 \pm 0.26$ | $20.1 \pm 0.037$ | $0.687 \pm 0.00132$ | $0.143 \pm 0.00049$ |
| TECO-ConvSSM (ablation) | 175M | $44.3 \pm 2.69$ | $21.0 \pm 0.106$ | $0.691 \pm 0.00004$ | $0.010 \pm 0.00267$ |
| TECO-ConvS5 | 175M | $\underline{31.2 \pm 0.23}$ | $\mathbf{23.8 \pm 0.056}$ | $\mathbf{0.803 \pm 0.0020}$ | $\underline{0.085 \pm 0.00179}$ |

Table 9: Full results for ablation of ConvS5 convolutional ablations for DMLab long-range benchmark dataset [13]. To make parameter counts comparable for different configurations, when possible, we adjust parameters in the activation blocks, e.g. increasing the size of the ResNet convolution kernels or increasing the features of the GLU activation. With these adjustments, the models also have similar training speeds. Note the last entry is the S5 run which serves as an additional ablation of the convolutional structure.

| $\mathcal{B}$ kernel | $\mathcal{C}$ kernel | Activation | Params | DMLab | | | | Train Step Time (s) ↓ |
| --- | --- | --- | --- | --- | --- | --- | --- | --- |
| | | | | FVD ↓ | PSNR ↑ | SSIM ↑ | LPIPS ↓ | |
| $3 \times 3$ | $3 \times 3$ | ResNet | 100M | $\mathbf{66.6 \pm 4.81}$ | $\mathbf{23.2 \pm 0.053}$ | $\mathbf{0.769 \pm 0.00043}$ | $\mathbf{0.079 \pm 0.00073}$ | 2.31 |
| $1 \times 1$ | $3 \times 3$ | ResNet | 85M | $67.5 \pm 0.73$ | $22.8 \pm 0.041$ | $0.756 \pm 0.00039$ | $0.085 \pm 0.00124$ | 2.52 |
| $1 \times 1$ | $1 \times 1$ | ResNet | 100M | $81.1 \pm 6.06$ | $23.0 \pm 0.074$ | $0.767 \pm 0.00186$ | $0.083 \pm 0.00139$ | 2.21 |
| $3 \times 3$ | $3 \times 3$ | GLU | 71M | $96.1 \pm 5.20$ | $22.7 \pm 0.157$ | $0.762 \pm 0.00363$ | $0.088 \pm 0.00287$ | 2.75 |
| $1 \times 1$ | $5 \times 5$ | GLU | 83M | $89.1 \pm 0.61$ | $21.5 \pm 0.072$ | $0.713 \pm 0.00290$ | $0.106 \pm 0.00289$ | 2.58 |
| $1 \times 1$ | $1 \times 1$ | GLU | 30M | $187 \pm 2.77$ | $21.0 \pm 0.064$ | $0.689 \pm 0.00007$ | $0.112 \pm 0.00183$ | 1.73 |
| - | - | GLU | 140M | $221 \pm 13.1$ | $19.3 \pm 0.128$ | $0.641 \pm 0.00400$ | $0.162 \pm 0.04510$ | 1.34 |

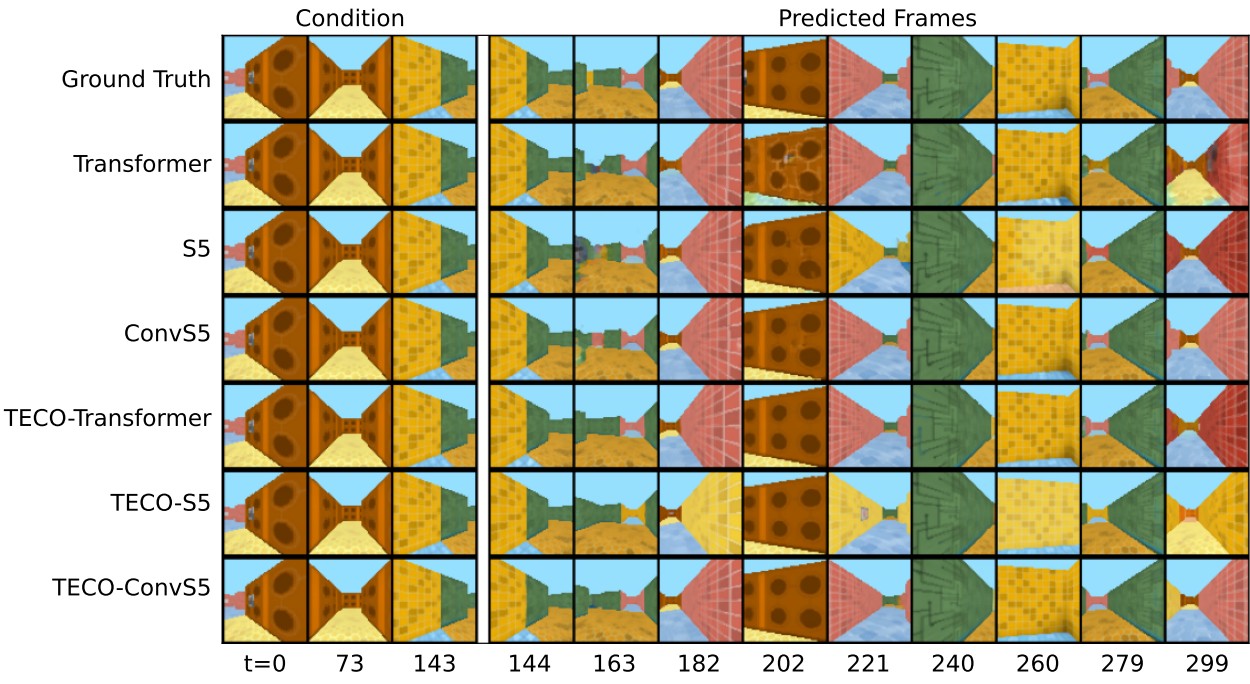

(a) 156 frames generated conditioned on 144 (action-conditioned).

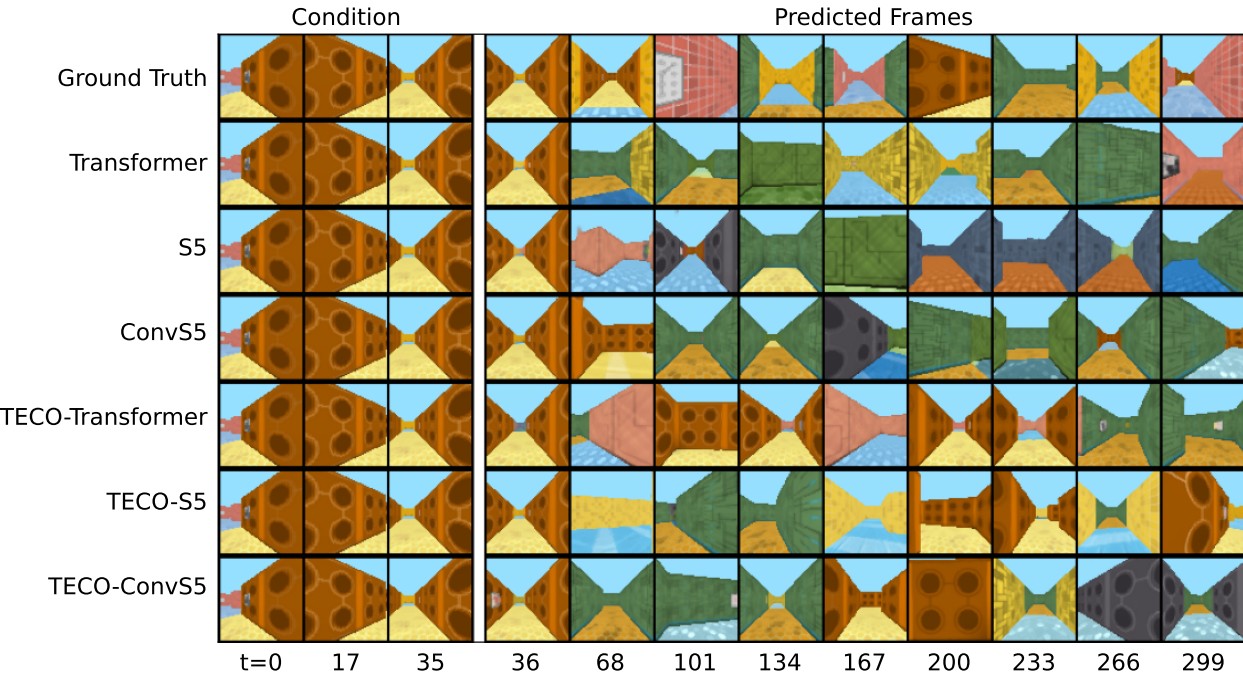

(b) 264 frames generated conditioned on 36 (**no** action-conditioning).

Figure 4: DMLab Samples

Condition                    Predicted Frames

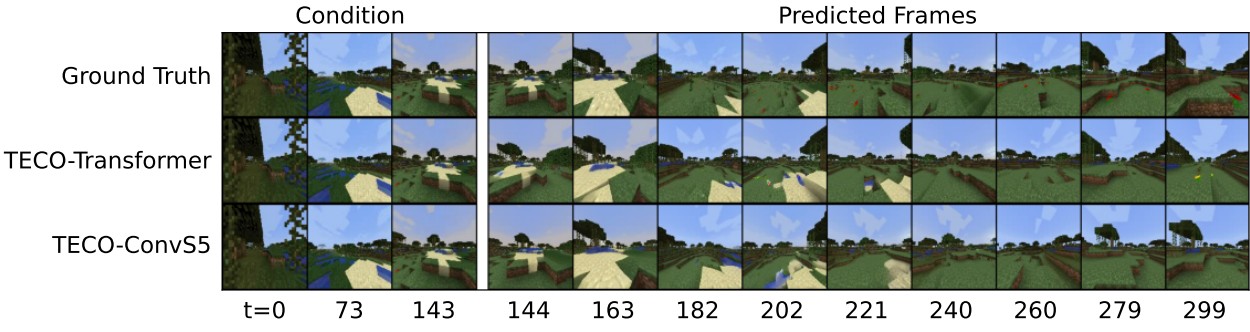

t=0    73    143    144    163    182    202    221    240    260    279    299

(a) 156 frames generated conditioned on 144 (action-conditioned).

Condition                    Predicted Frames

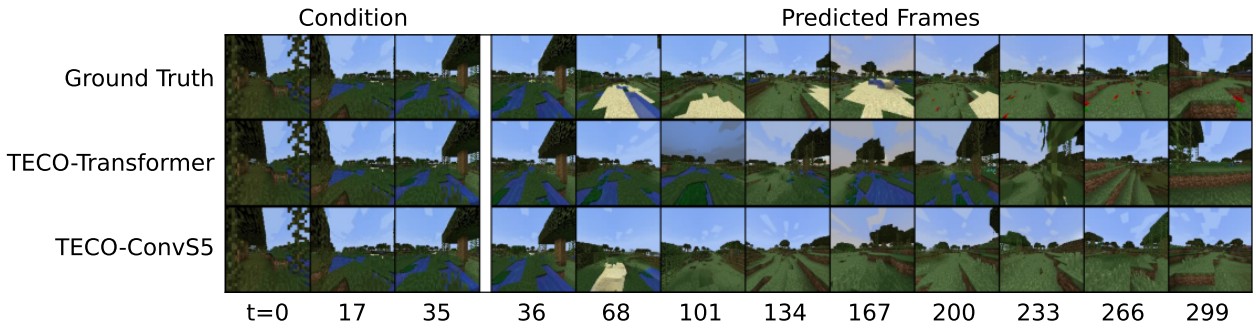

t=0    17    35    36    68    101    134    167    200    233    266    299

(b) 264 frames generated conditioned on 36 (action-conditioned).

Figure 5: Minecraft Samples

Condition                    Predicted Frames

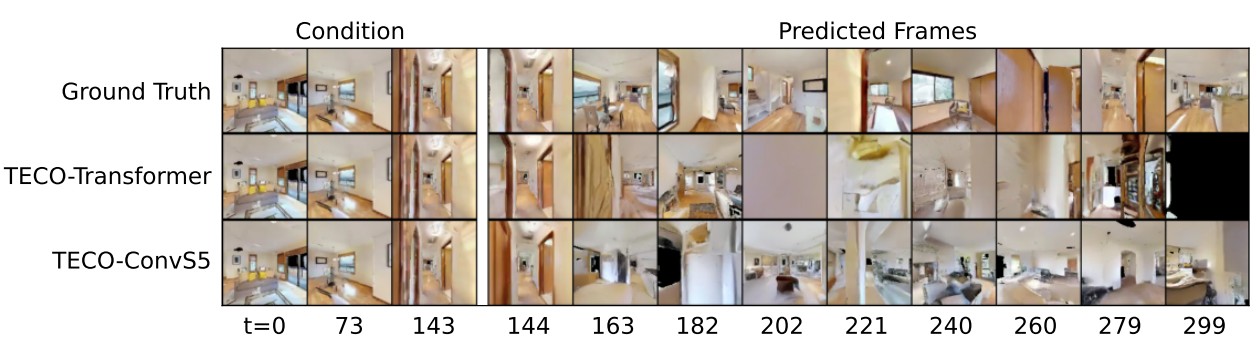

t=0    73    143    144    163    182    202    221    240    260    279    299

(a) 156 frames generated conditioned on 144 (action-conditioned).

Condition                    Predicted Frames

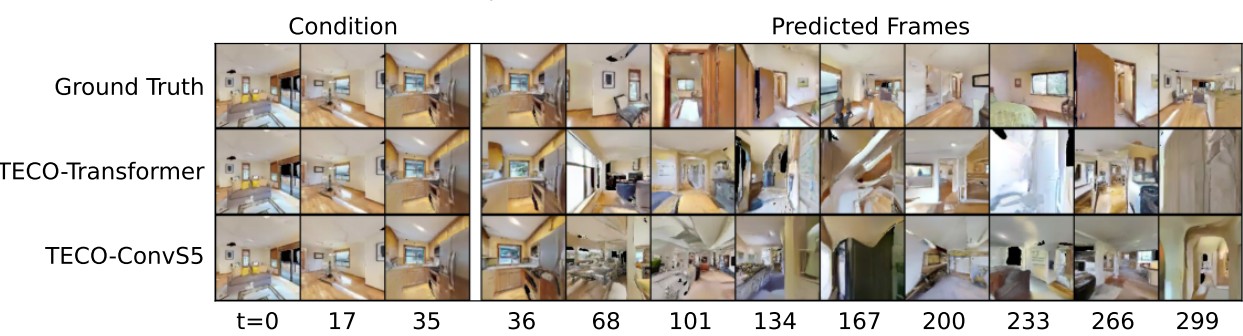

t=0    17    35    36    68    101    134    167    200    233    266    299

(b) 264 frames generated conditioned on 36 (**no** action-conditioning).

Figure 6: Habitat Samples

Table 10: Full results on the Minecraft and Habitat long-range benchmark datasets [13]. Results from Yan et al. [13] are indicated with ∗. Note that Yan et al. [13] did not evaluate FitVid or CW-VAE on Habitat due to cost.

| | | Minecraft | | | |
|---|---|---|---|---|---|
| Method | Params | FVD ↓ | PSNR ↑ | SSIM ↑ | LPIPS ↓ |
| FitVid* | 176M | $956 \pm 15.8$ | $13.0 \pm 0.0089$ | $0.343 \pm 0.00380$ | $0.519 \pm 0.00367$ |
| CW-VAE* | 140M | $397 \pm 15.5$ | $13.4 \pm 0.0610$ | $0.338 \pm 0.00274$ | $0.441 \pm 0.00367$ |
| Perceiver AR* | 166M | $\underline{76.3 \pm 1.72}$ | $13.2 \pm 0.0711$ | $0.323 \pm 0.00336$ | $0.441 \pm 0.00207$ |
| Latent FDM* | 33M | $167 \pm 6.26$ | $13.4 \pm 0.0904$ | $0.349 \pm 0.00327$ | $0.429 \pm 0.00284$ |
| TECO-Transformer* | 274M | $116 \pm 5.08$ | $\mathbf{15.4 \pm 0.0603}$ | $\mathbf{0.381 \pm 0.00192}$ | $\mathbf{0.340 \pm 0.00264}$ |
| TECO-ConvS5 | 214M | $\mathbf{70.7 \pm 3.05}$ | $\underline{14.8 \pm 0.0984}$ | $\underline{0.374 \pm 0.00414}$ | $\underline{0.355 \pm 0.00467}$ |

| | | Habitat | | | |
|---|---|---|---|---|---|
| Method | Params | FVD ↓ | PSNR ↑ | SSIM ↑ | LPIPS ↓ |
| Perceiver AR* | 200M | $164 \pm 12.6$ | $\underline{12.8 \pm 0.0423}$ | $\mathbf{0.405 \pm 0.00248}$ | $0.676 \pm 0.00282$ |
| Latent FDM* | 87M | $433 \pm 2.67$ | $12.5 \pm 0.0121$ | $0.311 \pm 0.00083$ | $\mathbf{0.582 \pm 0.00049}$ |
| TECO-Transformer* | 386M | $\mathbf{76.3 \pm 1.72}$ | $\underline{12.8 \pm 0.0139}$ | $0.363 \pm 0.00122$ | $\underline{0.604 \pm 0.00451}$ |
| TECO-ConvS5 | 351M | $\underline{95.1 \pm 3.74}$ | $\mathbf{12.9 \pm 0.212}$ | $\underline{0.390 \pm 0.01238}$ | $0.632 \pm 0.00823$ |

Table 11: Model runtime comparison for 3D Environment results in Tables 8-10. The implementations of the baselines FitVid, CW-VAE, Perceiver AR and Latent FDM used in the TECO work [13] are not publicly available in the TECO repository, so we were unable to include direct runtime comparisons for those methods.

| | DMLab | |
|---|---|---|
| Method | Train Step Time (s) ↓ | Sampling Speed (frames/s) |
| Transformer | $\mathbf{1.25}\ (\mathbf{1.0}\times)$ | $9.1\ (1.0\times)$ |
| Performer | $1.25\ (1.0\times)$ | $7.6(0.8\times)$ |
| S5 | $\underline{1.34\ (1.1\times)}$ | $28\ (3.1\times)$ |
| ConvS5 | $2.31\ (1.8\times)$ | $\mathbf{56}\ (\mathbf{6.2}\times)$ |
| TECO-Transformer | $\mathbf{0.75}\ (\mathbf{0.6}\times)$ | $16\ (1.8\times)$ |
| TECO-S5 | $\underline{0.81\ (0.7\times)}$ | $\mathbf{21}\ (\mathbf{2.3}\times)$ |
| TECO-ConvS5 | $1.17\ (0.9\times)$ | $18\ (2.0\times)$ |

| | Minecraft | |
|---|---|---|
| Method | Train Step Time (s) ↓ | Sampling Speed (frames/s) |
| TECO-Transformer | $\mathbf{1.91}\ (\mathbf{1.0}\times)$ | $8.1\ (1.0\times)$ |
| TECO-ConvS5 | $\underline{2.53\ (1.3\times)}$ | $\mathbf{14}\ (\mathbf{1.7}\times)$ |

| | Habitat | |
|---|---|---|
| Method | Train Step Time (s) ↓ | Sampling Speed (frames/s) |
| TECO-Transformer | $\mathbf{2.71}\ (\mathbf{1.0}\times)$ | $6.8\ (1.0\times)$ |
| TECO-ConvS5 | $\underline{3.10\ (1.1\times)}$ | $\mathbf{11}\ (\mathbf{1.6}\times)$ |

# D Experiment Configurations

Our codebase modifies the TECO codebase from Yan et al. [13] and we reuse their core Transformer and TECO framework implementations. More architectural details and dataset-specific details are described below.

## D.1 Spatiotemporal Sequence Model Architectures

ConvS5, ConvLSTM and S5 models are formed by stacking multiple ConvS5, ConvLSTM or S5 layers, respectively. For each of these models, layer normalization [107] with a post-norm setup is used along with residual connections. For the Transformer, we use the Transformer implementation from Yan et al. [13] which consists of a stack of multi-head attention layers.

ConvS5 and ConvLSTM are applied directly to sequences of frames of shape [sequence length, latent height, latent width, latent features], where the original data has been convolved to a latent resolution and latent number of features. Since S5 and Transformer act on vector-valued sequences, these models require an additional downsampling convolution operation to project the latent frames into a token and an upsampling transposed convolution operation to project the Transformer backbone output tokens back into latent frames. We use the same sequence of compression operations for this as in Yan et al. [13]. The Encoder and Decoder referred to for all models in the hyperparameter tables below consist of ResNet Blocks with $3 \times 3$ kernels as implemented in Yan et al. [13].

## D.2 Evaluation Metrics

We follow Yan et al. [13] and evaluate methods by computing Fréchet Video Distance (FVD) [108], peak signal-to-noise ratio (PSNR), structural similarity index measure [109] and Learned Perceptual Image Patch Similarity (LPIPS) [110] between sampled trajectories and ground truth trajectories. See Yan et al. [13] for a more in-depth discussion of the use of these metrics for the 3D environment benchmarks.

## D.3 Compute

All models were trained with 32GB NVIDIA V100 GPUs. For Moving-MNIST, models were trained with 8 V100s. For all other experiments, models were trained with 16 V100s. We list V100 days in the hyperparameters, which denotes the number of days it would take to train on a single V100.

## D.4 Moving-MNIST

All models were trained to minimize L1+L2 loss over the frames directly in pixel space, as in Su et al. [84]. We trained models on 300 frames. We then repeated the experiment and trained models on 600 frames. For ConvS5 and ConvLSTM, we fixed the hidden dimensions (layer input/output features) and state sizes to be 256, and we swept over the following learning rates $[1 \times 10^{-4}, 5 \times 10^{-4}, 1 \times 10^{-3}]$ and chose the best model. For Transformer, we swept over model size, considering hidden dimensions of [512, 2014] and learning rates $[1 \times 10^{-4}, 5 \times 10^{-4}, 1 \times 10^{-3}]$ and chose the best model. We also observed better performance for the Transformer by convolving frames down to an $8 \times 8$ latent resolution (rather than the $16 \times 16$ used by ConvS5 and ConvLSTM) before downsampling to a token. All other relevant training parameters were kept the same between the three methods. See Tables 12-14 for detailed experiment configurations.

Each model was evaluated by collecting 1024 trajectories using the following procedure: condition on 100 frames from the ground truth test set, then generate forward 1200 frames. These samples were compared with the ground truth to compute FVD, PSNR, SSIM and LPIPS.

The ConvSSM ablation was performed using the exact settings as ConvS5, except the state kernel was initialized with a Gaussian and we swept over the following learning rates $[1 \times 10^{-4}, 5 \times 10^{-4}, 1 \times 10^{-3}]$.

Table 12: Experiment Configuration for ConvS5 on Moving-MNIST experiments

|  | Hyperparameters | Moving-MNIST-300 | Moving-MNIST-600 |
|---|---|---|---|
|  | V100 Days | 25 | 50 |
|  | Params | 20M | 20M |
|  | Input Resolution | $64 \times 64$ | $64 \times 64$ |
|  | Latent Resolution | $16 \times 16$ | $16 \times 16$ |
|  | Batch Size | 8 | 8 |
|  | Sequence Length | 300 | 600 |
|  | LR | $1 \times 10^{-3}$ | $1 \times 10^{-3}$ |
|  | LR Schedule | cosine | cosine |
|  | Warmup Steps | 5k | 5k |
|  | Max Training Steps | 300K | 300K |
|  | Weight Decay | $1 \times 10^{-5}$ | $1 \times 10^{-5}$ |
| Encoder | Depths | 64, 128, 256 | 64, 128, 256 |
|  | Blocks | 1 | 1 |
| Decoder | Depths | 64, 128, 256 | 64, 128, 256 |
|  | Blocks | 1 | 1 |
| ConvS5 | Hidden Dim ($U$) | 256 | 256 |
|  | State Size ($P$) | 256 | 256 |
|  | $\mathcal{B}$ Kernel Size | $3 \times 3$ | $3 \times 3$ |
|  | $\mathcal{C}$ Kernel Size | $3 \times 3$ | $3 \times 3$ |
|  | Layers | 8 | 8 |
|  | Dropout | 0 | 0 |
|  | Activation | ResNet | ResNet |

Table 13: Experiment Configuration for ConvLSTM on Moving-MNIST experiments

|  | Hyperparameters | Moving-MNIST-300 | Moving-MNIST-600 |
|---|---|---|---|
|  | V100 Days | 75 | 150 |
|  | Params | 20M | 20M |
|  | Input Resolution | $64 \times 64$ | $64 \times 64$ |
|  | Latent Resolution | $16 \times 16$ | $16 \times 16$ |
|  | Batch Size | 8 | 8 |
|  | Sequence Length | 300 | 600 |
|  | LR | $5 \times 10^{-4}$ | $5 \times 10^{-4}$ |
|  | LR Schedule | cosine | cosine |
|  | Warmup Steps | 5k | 5k |
|  | Max Training Steps | 300K | 300K |
|  | Weight Decay | $1 \times 10^{-5}$ | $1 \times 10^{-5}$ |
| Encoder | Depths | 64, 128, 256 | 64, 128, 256 |
|  | Blocks | 1 | 1 |
| Decoder | Depths | 64, 128, 256 | 64, 128, 256 |
|  | Blocks | 1 | 1 |
| ConvLSTM | Hidden Dim | 256 | 256 |
|  | State Size | 256 | 256 |
|  | Kernel Size | $3 \times 3$ | $3 \times 3$ |
|  | Layers | 8 | 8 |
|  | Dropout | 0 | 0 |

Table 14: Experiment Configuration for Transformer on Moving-MNIST experiments

| Hyperparameters | | Moving-MNIST-300 | Moving-MNIST-600 |
|---|---|---|---|
| | V100 Days | 25 | 50 |
| | Params | 164M | 164M |
| | Input Resolution | $64 \times 64$ | $64 \times 64$ |
| | Latent Resolution | $8 \times 8$ | $8 \times 8$ |
| | Batch Size | 8 | 8 |
| | Sequence Length | 300 | 600 |
| | LR | $5 \times 10^{-4}$ | $1 \times 10^{-4}$ |
| | LR Schedule | cosine | cosine |
| | Warmup Steps | 5k | 5k |
| | Max Training Steps | 300K | 300K |
| | Weight Decay | $1 \times 10^{-5}$ | $1 \times 10^{-5}$ |
| Encoder | Depths | 64, 128, 256, 512 | 64, 128, 256, 512 |
| | Blocks | 1 | 1 |
| Decoder | Depths | 64, 128, 256, 512 | 64, 128, 256, 512 |
| | Blocks | 1 | 1 |
| | Downsample Factor | 8 | 8 |
| | Hidden Dim | 1024 | 1024 |
| Temporal | Feedforward Dim | 4096 | 4096 |
| Transformer | Heads | 16 | 16 |
| | Layers | 8 | 8 |
| | Dropout | 0 | 0 |

### D.5 Long-Range 3D Environment Benchmarks

We follow the procedures from Yan et al. [13] and train models on the same pre-trained vector-quantized (VQ) $16 \times 16$ codes used by the baselines evaluated in that work. Models were trained to optimize a cross-entropy reconstruction loss between the predictions and true VQ codes. The evaluation of DMLab and Habitat involves both an action-conditioned and unconditioned setting. Therefore, as in Yan et al. [13], the actions were randomly dropped out half the time during training on these datasets.

After training, we follow the procedure from Yan et al. [13] for evaluation in two different settings. The first setting involves computing PSNR, SSIM and LPIPS from 1024 samples generated by conditioning on the first 144 frames and then generating the next 156 frames while providing the model with past and future actions. The second setting does not provide actions as input (with the exception of Minecraft, which also provides actions in this setting). It involves computing FVD using 1024 samples generated by conditioning on the first 36 frames and then predicting the remaining 264 frames.

All sequence models we trained used the same number of layers as the Transformer used in the TECO-Transformer trained by Yan et al. [13]. In addition, the TECO-Transformer, TECO-S5 and TECO-ConvS5 models we trained used the exact encoder/decoder configuration and MaskGit configuration as in Yan et al. [13]. The Transformer, S5 and ConvS5 models we trained without the TECO framework were all trained using the same encoder/decoder configuration. See Tables 15-21 for more detailed experimental configuration details. See dataset-specific paragraphs below for hyperparameter tuning information.

**TECO Training Framework** Yan et al. [13] proposed the TECO training framework to train Transformers on long video data. For some of our experiments, we use ConvS5 layers and S5 layers as a drop-in replacement for the Transformer in this framework. We refer the reader to Yan et al. [13] for full details. Briefly, given the original VQ codes, TECO trains an additional encoder/decoder that compresses the frames to a lower latent resolution (e.g., from $16 \times 16$ to $8 \times 8$) by training an additional encoder/decoder with a codebook loss, $\mathcal{L}_{\mathrm{VQ}}$. In addition, a MaskGit [98] dynamics prior loss, $\mathcal{L}_{\mathrm{prior}}$, is used for the latent transitions. The sequence model (e.g. Transformer, S5, ConvS5) takes the latent frames (compressed into tokens in the case of Transformer and S5) and produces an output which is used along with the latents by the decoder to produce predictions and a reconstruction loss, $\mathcal{L}_{\mathrm{recon}}$. Models are trained to minimize the following total loss:

$$\mathcal{L}_{\mathrm{TECO}} = \mathcal{L}_{\mathrm{VQ}} + \mathcal{L}_{\mathrm{recon}} + \mathcal{L}_{\mathrm{prior}}. \tag{32}$$

In addition, TECO includes the use of DropLoss [13], which drops out a percentage of random timesteps that are not decoded and therefore do not require computing the expensive $\mathcal{L}_{\mathrm{recon}}$ and $\mathcal{L}_{\mathrm{prior}}$ terms.

**DMLab** As mentioned above, the actions were randomly dropped out of sequences half the time (due to the two evaluation scenarios, action-conditioned and unconditioned). We observed that for DMLab, when provided past and future actions, models converged faster using the simple masking strategy discussed in Gu et al. [19] that masks the future inputs rather than feeding the predicted inputs (or true inputs during training) autoregressively. Therefore we trained all models (Transformer, Performer, S5, ConvS5, Teco-Transformer, TECO-S5, TECO-ConvS5) by using this strategy when the actions were provided, and using the autoregressive strategy when actions were not provided. We observed this significantly improved the LPIPS of the Transformer baselines. Note, in pilot runs for Minecraft and Habitat, we observed this strategy led to lower-quality frames and did not use it for the reported results for those datasets.

We trained each model, Transformer, Performer, S5, ConvS5, Teco-Transformer, TECO-S5, TECO-ConvS5, with three different learning rates $[1 \times 10^{-4}, 5 \times 10^{-4}, 1 \times 10^{-3}]$ and selected the best run for each model. See Tables 15-20 for more experiment configuration details.

The TECO-ConvSSM ablation used the exact same settings as TECO-ConvS5, except the state kernel was initialized with a random Gaussian and a lower learning rate of $1 \times 10^{-5}$ was required for stable training.

Table 15: Experiment Configuration for ConvS5 on DMLab

|  | Hyperparameters | DMLab |
|---|---|---|
|  | V100 Days | 150 |
|  | Params | 101M |
|  | Input Resolution | $64 \times 64$ |
|  | Latent Resolution | $16 \times 16$ |
|  | Batch Size | 16 |
|  | Sequence Length | 300 |
|  | LR | $5 \times 10^{-4}$ |
|  | LR Schedule | cosine |
|  | Warmup Steps | 5k |
|  | Max Training Steps | 500K |
|  | Weight Decay | $1 \times 10^{-5}$ |
| Encoder | Depths | 256 |
|  | Blocks | 1 |
| Decoder | Depths | 256 |
|  | Blocks | 4 |
| ConvS5 | Hidden Dim ($U$) | 512 |
|  | State Size ($P$) | 512 |
|  | $\mathcal{B}$ Kernel Size | $3 \times 3$ |
|  | $\mathcal{C}$ Kernel Size | $3 \times 3$ |
|  | Layers | 8 |
|  | Dropout | 0 |
|  | Activation | ResNet |

Table 16: Experiment Configuration for S5 on DMLab

|  | Hyperparameters | DMLab |
|---|---|---|
|  | V100 Days | 125 |
|  | Params | 140M |
|  | Input Resolution | $64 \times 64$ |
|  | Latent Resolution | $16 \times 16$ |
|  | Batch Size | 16 |
|  | Sequence Length | 300 |
|  | LR | $1 \times 10^{-3}$ |
|  | LR Schedule | cosine |
|  | Warmup Steps | 5k |
|  | Max Training Steps | 500K |
|  | Weight Decay | $1 \times 10^{-5}$ |
| Encoder | Depths | 256 |
|  | Blocks | 1 |
| Decoder | Depths | 256 |
|  | Blocks | 4 |
| S5 | Downsample Factor | 16 |
|  | Hidden Dim ($U$) | 1024 |
|  | State Size ($P$) | 1024 |
|  | Layers | 8 |
|  | Dropout | 0 |
|  | Activation | GLU (half) |

Table 17: Experiment Configuration for Transformer on DMLab

| Hyperparameters | | DMLab |
|---|---|---|
| | V100 Days | 125 |
| | Params | 152M |
| | Input Resolution | $64 \times 64$ |
| | Latent Resolution | $16 \times 16$ |
| | Batch Size | 16 |
| | Sequence Length | 300 |
| | LR | $5 \times 10^{-4}$ |
| | LR Schedule | cosine |
| | Warmup Steps | 5k |
| | Max Training Steps | 500K |
| | Weight Decay | $1 \times 10^{-5}$ |
| Encoder | Depths | 256 |
| | Blocks | 1 |
| Decoder | Depths | 256 |
| | Blocks | 4 |
| | Downsample Factor | 16 |
| | Hidden Dim | 512 |
| Temporal | Feedforward Dim | 2048 |
| Transformer | Heads | 16 |
| | Layers | 8 |
| | Dropout | 0 |

Table 18: Experiment Configuration for TECO-ConvS5 on DMLab

| Hyperparameters | DMLab | |
|---|---|---|
| | V100 Days | 110 |
| | Params | 175M |
| | Input Resolution | $64 \times 64$ |
| | Latent Resolution | $8 \times 8$ |
| | Batch Size | 16 |
| | Sequence Length | 300 |
| | LR | $5 \times 10^{-4}$ |
| | LR Schedule | cosine |
| | Warmup Steps | 5k |
| | Max Training Steps | 500K |
| | Weight Decay | $1 \times 10^{-5}$ |
| | DropLoss Rate | 0.9 |
| Encoder | Depths | 256, 512 |
| | Blocks | 2 |
| Codebook | Size | 1024 |
| | Embedding Dim | 32 |
| Decoder | Depths | 256, 512 |
| | Blocks | 4 |
| ConvS5 | Hidden Dim ($U$) | 512 |
| | State Size ($P$) | 1024 |
| | $\mathcal{B}$ Kernel Size | $3 \times 3$ |
| | $\mathcal{C}$ Kernel Size | $3 \times 3$ |
| | Layers | 8 |
| | Dropout | 0 |
| | Activation | ResNet |
| MaskGit | Mask Schedule | cosine |
| | Hidden Dim | 512 |
| | Feedforward Dim | 2048 |
| | Heads | 8 |
| | Layers | 8 |
| | Dropout | 0 |

Table 19: Experiment Configuration for TECO-S5 on DMLab

| Hyperparameters | DMLab | |
|---|---|---|
| | V100 Days | 80 |
| | Params | 180M |
| | Input Resolution | $64 \times 64$ |
| | Latent Resolution | $8 \times 8$ |
| | Batch Size | 16 |
| | Sequence Length | 300 |
| | LR | $1 \times 10^{-3}$ |
| | LR Schedule | cosine |
| | Warmup Steps | 5k |
| | Max Training Steps | 500K |
| | Weight Decay | $1 \times 10^{-5}$ |
| | DropLoss Rate | 0.9 |
| Encoder | Depths | 256, 512 |
| | Blocks | 2 |
| Codebook | Size | 1024 |
| | Embedding Dim | 32 |
| Decoder | Depths | 256, 512 |
| | Blocks | 4 |
| S5 | Downsample Factor | 8 |
| | Hidden Dim ($U$) | 2048 |
| | State Size ($P$) | 2048 |
| | Layers | 8 |
| | Dropout | 0 |
| | Activation | GLU (half) |
| MaskGit | Mask Schedule | cosine |
| | Hidden Dim | 512 |
| | Feedforward Dim | 2048 |
| | Heads | 8 |
| | Layers | 8 |
| | Dropout | 0 |

Table 20: Experiment Configuration for TECO-Transformer on DMLab

| Hyperparameters | DMLab | |
|---|---|---|
| | V100 Days | 80 |
| | Params | 173M |
| | Input Resolution | $64 \times 64$ |
| | Latent Resolution | $8 \times 8$ |
| | Batch Size | 16 |
| | Sequence Length | 300 |
| | LR | $1 \times 10^{-4}$ |
| | LR Schedule | cosine |
| | Warmup Steps | 5k |
| | Max Training Steps | 500K |
| | Weight Decay | $1 \times 10^{-5}$ |
| | DropLoss Rate | 0.9 |
| Encoder | Depths | 256, 512 |
| | Blocks | 2 |
| Codebook | Size | 1024 |
| | Embedding Dim | 32 |
| Decoder | Depths | 256, 512 |
| | Blocks | 4 |
| Temporal Transformer | Downsample Factor | 8 |
| | Hidden Dim | 1024 |
| | Feedforward Dim | 4096 |
| | Heads | 16 |
| | Layers | 8 |
| | Dropout | 0 |
| MaskGit | Mask Schedule | cosine |
| | Hidden Dim | 512 |
| | Feedforward Dim | 2048 |
| | Heads | 8 |
| | Layers | 8 |
| | Dropout | 0 |

**Minecraft and Habitat**  For Minecraft and Habitat, we only trained TECO-ConvS5 due to the costs of training on these datasets. See dataset details in Appendix E and reported compute costs in Yan et al. [13]. For Minecraft, we evaluated two different learning rates $[1 \times 10^{-4}, 5 \times 10^{-4}]$ and chose the best. For Habitat, we only performed one run with no further tuning. See Table 21 for further experiment configuration details.

Table 21: Experiment Configuration for TECO-ConvS5 on Minecraft and Habitat

| | Hyperparameters | Minecraft | Habitat |
|---|---|---|---|
| | V100 Days | 470 | 575 |
| | Params | 214M | 351M |
| | Input Resolution | $128 \times 128$ | $128 \times 128$ |
| | Latent Resolution | $8 \times 8$ | $8 \times 8$ |
| | Batch Size | 16 | 16 |
| | Sequence Length | 300 | 300 |
| | LR | $5 \times 10^{-4}$ | $1 \times 10^{-4}$ |
| | LR Schedule | cosine | cosine |
| | Warmup Steps | 5k | 5k |
| | Max Training Steps | 1M | 1M |
| | DropLoss Rate | 0.9 | 0.9 |
| Encoder | Depths | 256, 512 | 256, 512 |
| | Blocks | 4 | 4 |
| Codebook | Size | 1024 | 1024 |
| | Embedding Dim | 32 | 32 |
| Decoder | Depths | 256, 512 | 256, 512 |
| | Blocks | 8 | 8 |
| ConvS5 | Hidden Dim ($U$) | 512 | 512 |
| | State Size ($P$) | 512 | 512 |
| | $\mathcal{B}$ Kernel Size | $3 \times 3$ | $3 \times 3$ |
| | $\mathcal{C}$ Kernel Size | $3 \times 3$ | $3 \times 3$ |
| | Layers | 12 | 8 |
| | Dropout | 0 | 0 |
| | Activation | ResNet | ResNet |
| MaskGit | Mask Schedule | cosine | cosine |
| | Hidden Dim | 768 | 1024 |
| | Feedforward Dim | 3072 | 4096 |
| | Heads | 12 | 16 |
| | Layers | 6 | 16 |
| | Dropout | 0 | 0 |

# E Datasets

## E.1 Moving-MNIST

The Moving-MNIST [54] dataset is generated by moving two $28 \times 28$ size MNIST digits from the MNIST dataset [111] inside a $64 \times 64$ black background. The digits begin at a random initial location, and move with constant velocity, bouncing when they reach the boundary. For each of the sequence lengths we consider, 300 and 600, we follow Wang et al. [81] and Su et al. [84] and generate 10,000 sequences for training.

## E.2 DMLab

We use the DMLab long-range benchmark designed by Yan et al. [13] using the DeepMind Lab (DMLab) [99] simulator. The simulator generates random 3D mazes with random floor and wall textures. The benchmark consists of 40K action-conditioned, 300 frame videos at a $64 \times 64$ resolution. The videos are of an agent randomly navigating $7 \times 7$ mazes by choosing random points in the maze and navigating to them through the shortest path.

## E.3 Minecraft

We use the Minecraft [100] long-range benchmark designed by Yan et al. [13]. The game features 3D worlds that contain complex terrains such as hills, forests, rivers and lakes. The benchmark was constructed by collecting 200K action-conditioned 300 frame videos at a $128 \times 128$ resolution. The videos are in Minecraft's marsh biome and the agent iterates walking forward for a random number of steps and randomly rotating left or right. This results in parts of the scene going out of view and coming back into view later.

## E.4 Habitat

We use the Habitat long-range benchmark designed by Yan et al. [13] using the Habitat simulator [101]. The simulator renders trajectories using scans of real 3D scenes. Yan et al. [13] compiled 1400 indoor scans from HM3D [112], Matterport3D [113] and Gibson [114] to generate 200K action-conditioned, 300 frame videos with a $128 \times 128$ resolution. Yan et al. [13] used Habitat's in-built path traversal algorithm to construct action trajectories that move the agent between randomly sampled locations.

