# OpenReview forum: "Convolutional State Space Models for Long-Range Spatiotemporal Modeling"
_NeurIPS.cc/2023/Conference — NeurIPS 2023 poster_

### Official Review · Reviewer_xT6A · 2023-07-01

**Soundness:** 3 good
**Presentation:** 3 good
**Contribution:** 2 fair
**Rating:** 7
**Confidence:** 4

**Summary:**

This paper presents ConvS5, a convolutional state-space model that aims to model long-range spatiotemporal dependencies in video data. They extend the prior S5 model to operate in a convolutional state-space, and retain the long-range benefits of state-space models by using a point-wise convolutional kernel for diagonalizable and S5 HiPPO initialization. The resulting model can then be efficiently trained using parallel scans due to linearity in the recurrent mechanism. They show strong results on several complex long-range video benchmarks in Moving MNIST, DMLab, Minecraft, and Habitat.

**Strengths:**

- The paper is generally clear and well written
- Comprehensive experiments on a range of datasets focused on long-range understanding in videos, showing that their proposed method is better than or competitive with SOTA methods
- Experiments show that better scaling in sequence length allows for training on longer sequences, which results in better / more stable generation
- Faster inference speed compared to existing methods while retaining high generation fidelity

**Weaknesses:**

- Overall technical novelty is a little low, as the model itself is a straightforward extension of the S5 model to a convolutional setting.
- A primary concern I have about recurrent-based models for long-range understanding in video generation is limited capacity in the recurrent state, as these models are required to store all fine-grained details about the scene in the recurrent state (since potentially every part of the scenes will be visited / need to be reconstructed at a future timestep), whereas transformer-based models can dynamically retrieve from all prior states for each timestep. A rigorous study on this aspect would benefit the paper greatly, as this issue is more present in visually complex videos (minecraft -> habitat -> real videos), and could be a potential bottleneck to the scalability of these models.

**Questions:**

- For Moving MNIST, how would ConvS5 potentially compare against temporally hierarchical methods in modeling long-range dependencies (e.g. Clockwork-VAE)? Or would an analogous hierarchical version of ConvS5 be better than a hierarchical GRU model?
- How would ConvS5 compare for Minecraft / Habitat? I believe the primary difference is without the MaskGit / iterative decoding on the prior, so would ConvS5 struggle with more stochastic environments?
- What is the potential cause of the difference  in performance between FVD and PSNR / SSIM / LPIPS in Minecraft? Is FVD lower due to better frame fidelity of TECO-ConvS5, and TECO-transformer has better long-term consistency? Or perhaps other factors?
- Are generated video samples available anywhere to view? It is hard to evaluate temporal coherence in the rows of frames provided in the appendix.

**Limitations:**

Yes, the authors discuss limitations.

---

> ### Author Rebuttal · Authors · 2023-08-09
>
> We thank the reviewer for their positive feedback. We are glad the reviewer appreciated the comprehensive experiments as well as strong results.
>
> **--Re. technical novelty**: We respectfully disagree that the proposed approach is a straightforward extension of S5. Please see the Technical Contributions section of the General Response for a detailed explanation of the contributions required to ensure the convolutional recurrence could be efficiently parallelized and model long-range spatiotemporal dependencies.
>
> **--Re. recurrent vs attention architectures**: Thank you for the question. This is an interesting discussion. While it is true that recurrent methods are required to store the history in its recurrent states, this capacity can be increased by adding more hidden units or layers. On the other hand, the attention mechanisms have bounded context by their design which limits the amount of history they can retrieve. Therefore, it is not necessarily true that the transformer-based models can retrieve from all prior states for each timestep or that recurrent models are a bottleneck for scalability.
>
> In addition, recent architecture improvements have been proposed for SSMs in language modeling, such as multiplicative gating/routing operators (e.g. in H3 [1]) to allow for recall operations similar to those you have suggested. These ideas can be applicable to ConvSSMs as well. Another possibility is to apply chunked/bounded context attention mechanisms to the outputs of an SSM/ConvSSM as suggested in MEGA [2].
>
> However, the main focus of this paper was to develop the base ConvSSM architecture. Our evaluation on the long-range video benchmarks (the most challenging ones that currently exist) show that ConvS5 performs comparably or better than Transformers on these tasks. We believe this provides some evidence that this type of method is capable of scaling to more complex long-range datasets.
>
> We would be happy to include these discussions and potential modifications in the final version. Please let us know if the reviewer has suggestions for an additional study they would like to see in this regard.
>
> ### Questions:
>
> **--Re. comparing against temporally hierarchical methods:** Thank you for your suggestion!  We added the CW-VAE comparison to the MNIST table for models trained on 600 sequence length below. We will add a table for models trained on 300 sequence lengths for the final version (we could not finish both before the rebuttal deadline).  We found that ConvS5 outperforms CW-VAE in this setting across the metrics with both models having a similar number of parameters. This result indicates that ConvS5 itself compares well to temporally hierarchical methods in modeling long-range dependencies. However, we also agree that it could be interesting to combine temporally hierarchical methods with ConvSSM methods. We can include a discussion of this in the final version.
>
> **New CW-VAE baseline trained on 600 Frames:**
> |             |            |  100 &rarr; | 800         |              |            |  100 &rarr; | 1200        |              |
> |-------------|------------|------------:|-------------|--------------|------------|------------:|-------------|--------------|
> |             | FVD &darr; | PSNR &uarr; | SSIM &uarr; | LPIPS &darr; | FVD &darr; | PSNR &uarr; | SSIM &uarr; | LPIPS &darr; |
> | Transformer |   **42**   |     13.7    |    0.672    |     0.207    |     91     |     13.1    |    0.631    |     0.252    |
> | ConvLSTM    |     91     |     15.5    |    0.757    |     0.149    |     137    |     14.6    |    0.727    |     0.180    |
> | CW-VAE      |     93     |     12.5    |    0.599    |    0.268     |     109    |     12.4    |    0.590    |     0.280    |
> | ConvS5      |     47     |   **16.4**  |  **0.788**  |   **0.134**  |   **71**   |   **15.6**  |  **0.763**  |   **0.162**  |
>
> **--Re. convS5 comparison for Minecraft/Habitat**:
> We were not able to add the comparisons of (non-TECO) Transformer, S5, and ConvS5, because they are a lot more expensive to run than the TECO variants. Note that the size of the datasets and the model size used for these datasets are much bigger than DMLAB. However, we believe that the trend would be similar to the DMLAB non-TECO comparisons.
>
> Nevertheless, we may be able to try the comparison with smaller model sizes for these datasets if the reviewer would like to see the relative comparisons of the models for the final version. Please let us know.
>
> **--Re. performance difference between metrics on Minecraft**:
> We are not confident to make any strong conclusion about the difference in performance between FVD vs PSNR/SSIM/LPIPS. However, the gap between TECO-Transformer and TECO-ConvS5 is small for all metrics (especially relative to the baselines), and the visual differences in the prediction between these models were not significant. Our conclusion is that the performance of these models is comparable on this dataset, but our model has a faster sampling speed than TECO-Transformer (1.7x at this sequence generation length; note that the TECO-ConvS5 speed will stay constant for generating longer sequences while the TECO-Transformer speed will decrease).
>
> **--Re. link to videos**:
> In the appendix of the original submission (Line 746 in Appendix C), we included a link to an anonymized website that includes the video samples. We will move this link to the main paper to make this clear. Please see the link in the appendix. Note that we cannot include any external link in our response due to NeurIPS rebuttal policy.
>
> **We thank the reviewer again** for the time they have taken to review our paper and read this rebuttal. We hope we have addressed the reviewer's questions and the reviewer is willing to increase their score. Please let us know if we can provide additional clarification or information.
>
> **References:**
>  [1] Hungry Hungry Hippos: Towards Language Modeling with State Space Models. [2] Mega: Moving Average Equipped Gated Attention.

---

> > ### Comment · Reviewer_xT6A · 2023-08-10
> > **Response**
> >
> > Thank you for your rebuttal, and the extra experiments.
> >
> > **re: ConvS5 in Minecraft / Habitat**
> > Could the authors clarify on the ConvS5 architecture? Is it autoregressive per token, or autoregressive over time and predicts all tokens of the next image simulatenously, or something else?
> >
> > **re: novelty**
> > Please correct me if I am misunderstanding something, but the proposed method / architecture seems very similar to a simple baseline of a standard S5 layer that is vmaped across space with shared parameters for each spatial token - i.e. (`jax.vmap(s5_layer, axis=1)(x) # x is of shape [T, H * W, C]`) - equivalent to ConvS5 with $1\times 1$ kernels for both $A$ and $B$.  The only difference I see is that $B$ is replaced $k\times k$ kernel, motivated by a technical contribution of using the connection between convolutions and matrix multiplications applied to S5 / ConvS5.
> >
> > If my understanding above is correct, the contribution over just using S5 this way does not seem too significant, or is there large gain in the extra $B$ being some $k\times k$ instead of $1\times 1$ convolution?
> >
> > That being said, I would be happy to raise my score to a 6. My current main concern preventing further increase is the degree of impact / novelty of the contribution.

---

> > > ### Author Response · Authors · 2023-08-11
> > >
> > > We thank the reviewer for reading our rebuttal and increasing their score.
> > >
> > > --In these experiments, ConvS5 is autoregressive over time and predicts all tokens of the next image simultaneously. But it can also be plugged into other settings as you mentioned (e.g. autoregressive per token at each timestep, or using MaskGit as in the TECO version). Please let us know if we can provide further clarification regarding this.
> > >
> > > --The reviewer's understanding regarding the $B$ kernel is correct. To clarify, if all kernels (both $B$ and $C$) are replaced with $1 \times 1$ kernels, then it is equivalent to vmapping S5 across the (reshaped) pixels/tokens of the image. You can consider ConvS5 to be a generalization of simply vmapping S5. It allows for modulating the amount of convolution/weight sharing and shared spatial information that is fed to the dynamical system.
> > > However, our novel contribution is not only this generalization but also the development of parallel scans for convolutional recurrences and the parameterization scheme that retains SSM's properties. This provides a practical extension of S5 for spatiotemporal data. We are working to include additional ablations that address this.
> > >
> > > An additional contribution is developing the connection of ConvRNNs with recent SSM methods, and providing a parallelizable and efficient ConvRNN-like model which overcomes difficulties of both Transformers and traditional ConvRNNs. We think this can open up a new line of research to consider these alternative architectures.

---

> > > > ### Author Response · Authors · 2023-08-16
> > > > **Response to Reviewer xT6A's questions regarding kernel size**
> > > >
> > > > We would like to further address the reviewer's question: is there a large gain in $B$ and $C$ kernels being $k \times k$ instead of $1\times1$?
> > > >
> > > > The table below includes the ablation with different kernel sizes and nonlinearities on DMLAB. We ensure similar parameter counts for all models. These models have not fully converged yet (they are about halfway through and we evaluate all models at the same training step), but the results indicate that a larger kernel size significantly helps to improve the performance, especially for the FVD. We note that the rows with $1\times1$ kernels for both B and C are equivalent to vmapping an S5 layer which the reviewer mentioned. We will include the final results in the final version of the paper. These results reflect our observations during previous pilot experiments, though we did not fully quantify the gain. We thank the reviewer for highlighting this point as these results clearly improve the paper.
> > > >
> > > > We hope these results highlighting the value of the convolutional structure of ConvS5 fully address the reviewer's concerns and we hope the reviewer may further increase their score based on this clarification. Please let us know if we can provide further clarification.
> > > >
> > > > **Kernel size ablations on DMLAB**
> > > >
> > > > |         B Kernel        |         C Kernel        | Nonlinearity | FVD &darr; | PSNR &uarr; | SSIM &uarr; | LPIPS &darr; |
> > > > |:-----------------------:|:-----------------------:|:------------:|:----------:|:-----------:|:-----------:|:------------:|
> > > > | $\boldsymbol{3\times3}$ | $\boldsymbol{3\times3}$ |  **ResNet**  |  **77\***  |  **22.3\*** | **0.740\*** |  **0.090\*** |
> > > > |        $1\times1$       |        $3\times3$       |    ResNet    |     104    |     21.9    |    0.729    |     0.092    |
> > > > |        $1\times1$       |        $1\times1$       |    ResNet    |     207    |     21.9    |    0.726    |     0.094    |
> > > > |        $3\times3$       |        $3\times3$       |      GLU     |     122    |     20.7    |    0.677    |     0.118    |
> > > > |        $1\times1$       |        $5\times5$       |      GLU     |     144    |     20.4    |    0.670    |     0.126    |
> > > > |        $1\times1$       |        $1\times1$       |      GLU     |     381    |    19.95    |    0.644    |     0.134    |
> > > >
> > > > **bold**: final model, **\***: best number

---

> > > > > ### Comment · Reviewer_xT6A · 2023-08-16
> > > > > **Response**
> > > > >
> > > > > Thank you for running further experiments. I have a few quick clarifying questions:
> > > > >
> > > > > 1) How did you ensure similar parameter counts? i.e. between row 1 (B, C 3x3) and row 3 (B, C 1x1)?
> > > > > 2) What architecture was this? TECO-ConvS5 or ConvS5?
> > > > > 3) Nonlinearity here refers to the block in-between SSM layers?

---

> > > > > > ### Author Response · Authors · 2023-08-16
> > > > > >
> > > > > > Thank you for your questions:
> > > > > >
> > > > > > 1. Between row 1 and row 3 we increased the size of one of the convolutions in the ResNet block by using a $5\times5$ kernel.
> > > > > >
> > > > > > 2. This was the ConvS5 architecture (No TECO framework).
> > > > > >
> > > > > > 3. Yes, nonlinearity refers to the nonlinear connection/block between linear layers.
> > > > > >
> > > > > > Please let us know if we can provide any further clarifications.

---

> > > > > > > ### Comment · Reviewer_xT6A · 2023-08-18
> > > > > > > **Response**
> > > > > > >
> > > > > > > Thank you for the clarifications. After further consideration, I believe my concern about technical impact has been addressed (re: further experiments / ablations performed by the authors + other theoretical analysis regarding the ConvS5 formulation), and recommend acceptance (7). In the revised paper, I would encourage the authors to also include some measure of computational requirements for each row (e.g. wall clock time, GPU-hours, FLOPs, etc.) as additional columns in the table to better demonstrate the benefits of a convolutional state-space compared to just adding convolutions to a nonlinearity.

---

> > > > > > > > ### Author Response · Authors · 2023-08-18
> > > > > > > >
> > > > > > > > We thank the reviewer for carefully considering our rebuttal, asking thoughtful questions, and increasing our rating. We will include additional computational details in the table as suggested.

---

### Official Review · Reviewer_rhqo · 2023-07-01

**Soundness:** 3 good
**Presentation:** 3 good
**Contribution:** 2 fair
**Rating:** 5
**Confidence:** 3

**Summary:**

This paper proposes a new state space model for spatiotemporal modeling by introducing inductive bias of spatial locality. The core idea is to extend SSMs to ConvSSMs (just like extending FC-LSTM to ConvLSTM), which has a inherent convolutional structure. The new model also establishes an equivalence between the dynamics of particularly structured ConvSSMs and SSMs. Based on recent state space method S5, a model instantiation ConvS5 combines the stateful autoregressive generation with the ability to be parallelized across the sequence. A experimental evaluation shows that the proposed method captures spatiotemporal information better.

**Strengths:**

* The idea is simple but effective as it combines fast autoregressive generation and parallelized process.
* The proposed method achieves better performance than Transformer and ConvRNN on Moving-MNIST prediction task. It also strikes a balance on computational complexity between the two methods.

**Weaknesses:**

* The lack of different ConvSSM variants. The paper proposes ConvSSMs to address long-range spatiotemporal modeling. However, only a variant, ConvS5, is present and evaluated. It is not convincing to support the basic idea of ConvSSMs. More variants are needed.
* The experiments about comparison with Transformer and ConvLSTM could be more extensive. In Table 2, we see that the ConvS5 outperforms the other methods across the metrics. We notice that ConvLSTM achieves a slightly lower performance result. However, ConvLSTM shows a slightly better sampling speed than ConvS5. It is not sufficiently significant that ConvS5 has a better quality-speed tradeoff than ConvLSTM.
* There lacks explanation and experiments to the design of convolutional operator upon SSMs. The comparison results of whether convolution or not are missing.

**Questions:**

* The idea of parallel scan for SSMs is not new. This design is considerable overlap with S5, which also leverages parallel scans to efficiently compute the states. The authors are suggested to discuss more about this.
* In Table 3, ConvS5 shows a significant improvement than S5 without TECO. When trained using the TECO, ConvS5 only achieves a comparable result with S5. It makes me wonder if the proposed method is effective consistently on various settings (e.g. framework, SSMs backbone, etc).
* As in Weakness 3, I wonder if the proposed method is applicable to other SSMs. The authors are suggested to discuss more about this.

**Limitations:**

yes

---

> ### Author Rebuttal · Authors · 2023-08-09
>
> We thank the reviewer for the positive feedback. We are glad the reviewer appreciated the simplicity and effectiveness of the proposed method.
>
> --**Re. convSSM variants:** We present the general ideas of ConvSSMs (tensor-valued states, linear transitions and continuous-time parameterization) in Section 3.1, similar to how RNNs and ConvRNNs are general model classes with different variants. We note here that the general ConvSSM formulation of Section 3.1 could be run sequentially as is, similar to ConvRNNs. This could be considered a Vanilla ConvSSM variant. The downside of course is that this would be slow to train.
>
> We then work through the steps required to get a practical, effective, and modern version of a ConvSSM that allows efficient parallelization (Section 3.2), enables modeling long-range dependencies (Sections 3.3 and 3.4), while retaining the fast autoregressive generation abilities of ConvRNNs. The result is the ConvS5 method that we propose.
>
> Note that various recent extensions to SSMs such as data-dependent gating (H3 [1]) and data-dependent dynamics (Liquid SSMs [2]) could be applied to provide different extensions of ConvS5 (and thus different ConvSSM variants).  We can add a brief discussion of these potential extensions and variants in the discussion. However, our main focus in this work is to show the base ConvS5 architecture is effective.
>
> While the ConvS5 formulation we develop is simple and effective, we hope our work inspires others to consider the possibilities for this type of linear SSM and ConvRNN-inspired architecture, as well as other alternatives.
>
> If the reviewer has suggestions to improve the exposition of the general idea of ConvSSMs and the particular formulation of ConvS5, we are happy to incorporate these changes to make these distinctions more clear.
>
> --**Re. more comparison with Transformer and ConvLSTM:** We compare 3 aspects between these models. We believe they illustrate the main benefits of our method.
>
> 1. Performance: The experiments on Moving-MNIST show that
> ConvS5 is able to significantly outperform ConvLSTM across the metrics (cutting FVD nearly in half while also outperforming by a fair margin on PSNR/SSIM/LPIPS) when trained on the longer context (600 frames). This indicates that ConvS5 is able to take better advantage of the longer context and better capture the long-range dependencies than ConvLSTM.
>
> 2. Faster training: Because ConvS5 can be parallelized across the sequence, like Transformer, it trains much faster than the sequential ConvLSTM (>3X faster for these experiments, see Table 6 in the Appendix).
>
> 3. Fast autoregressive generation: Like ConvLSTM, ConvS5 has a much faster autoregressive generation speed than the Transformer. It is true that ConvLSTM had a slight edge (557x vs 427x faster than the Transformer) in these experiments, but the main point is that both methods are orders of magnitude faster than the Transformer.
>
> Also, ConvLSTM and ConvS5 are both stateful recurrent-based methods, so there is nothing that makes ConvLSTM inherently faster at autoregressive generation than ConvS5. Since both ConvS5 and ConvLSTM were much faster than the Transformer, and ConvS5 trains much faster than ConvLSTM, we did not spend time optimizing the speed of the generation process. Nonetheless, given the results from this experiment, there are likely many applications in which halving the FVD for a slightly slower inference speed (though still much faster than Transformer) would be worth it.
>
> We welcome suggestions from the reviewer for any additional comparisons to add here.
>
>
> --**Re. design and ablation of convolutions**: We discuss in detail the design of the convolution operators of ConvS5 in Section 3.2-3.4. Please let us know if we can provide further clarification on this point.
>
> In terms of ablating the convolutions, while we did not explicitly label it as an ablation, the S5 baselines in the DMLab experiments serve as this ablation since S5 is the closest possible variant to ConvS5 without the convolution structure. We will make a separate ablations table and discussion of these points to make this more clear. Please see the new ablations table and discussion in the General Response above.
>
> ### Questions:
>
> --**Re. the parallel scan design:** We propose a parallel scan for convolutional recurrences which is different from the one in S5. S5's parallel scan is for a linear recurrence while ConvS5's is for a convolutional recurrence.  There were several challenges for applying a parallel scan to a convolutional recurrence. We discuss the details of this in Section 3.2.  Section 2.3 explains the parallel scan used in S5. We would also be happy to incorporate suggestions the reviewer has on this point to help provide further clarification.
>
> --**Re. the effect of the TECO Framework:** We agree with the reviewer that the TECO framework provides S5 a large performance improvement on FVD (though still slightly below ConvS5). However we note that even with the TECO framework, ConvS5 still significantly outperforms S5 on the other long-range consistency metrics (PSNR/SSIM/LPIPS), so we respectfully disagree on the point that TECO-S5 achieves a comparable overall result to TECO-ConvS5. Therefore, ConvS5 performs well with and without the TECO framework, while S5 only performs well with the TECO framework. This would seem to suggest ConvS5 might be expected to generalize well to other settings since its performance on these experiments is not dependent on the framework.
>
> --**Re. other SSMs methods**: Please see the response of ConvSSM variants above.
>
> **We again thank the reviewer** for taking the time to review our paper. We hope we have answered the reviewer's questions and the reviewer is willing to increase their score. Please let us know if we can provide additional clarification or information.
>
> **References:** [1] Hungry Hungry Hippos: Towards Language Modeling with State Space Models [2] Liquid Structural State-Space Models

---

> ### Comment · Reviewer_rhqo · 2023-08-14
> **Response to authors**
>
> Thank the authors for providing the response and addressing my concerns. The motivation of this work looks much clearer to me now. My current main concern preventing further increase is the lack of different ConvSSM variants (not only the particular formulation of ConvS5). That being said, I would be happy to raise my score to 5.

---

> > ### Author Response · Authors · 2023-08-17
> >
> > We thank the reviewer for taking the time to read our rebuttal and for increasing their score!
> >
> > Re. ConvSSM variants:
> > Since the reviewer requested to introduce more ConvSSM variants, we can add another possible variant.
> > Here, motivated by the SSM to ConvSSM connection made in Section 3.3, we can consider a "ConvS4" variant related to S4 [3]. S4 uses single-input, single-output SSMs (one for each feature), so this would require applying a single-input, single-output SSM to every pixel and feature channel independently. From the ConvSSM point of view, this would require restricting the input/output kernels (B and C) and dynamics kernel A to $1\times1$ kernels and applying a different single-input, single-output ConvS4 to every channel independently. This stack of ConvS4s could be shared/convolved across each pixel of the frame.  For efficient sequence parallelization, FFTs would need to be used with this structure similar to S4 rather than using a parallel scan as used in ConvS5. In addition, an additional mixing operation would be required to mix the information of all the independent channels and pixels. It can be viewed as a depthwise-separable ConvSSM that could potentially reduce the number of parameters and operations. However, this approach has restricted kernel sizes, independent dynamical evolution of features, and a requirement of using FFTs and time-invariant dynamical systems.
> >
> > We can add a separate discussion section with "ConvSSM Variants" to the final paper which includes this ConvS4 variant, in addition to H3 [1] and Liquid SSM [2] variants that we mentioned in our rebuttal. We can also include an experiment for one of these variants.
> >
> > In addition to the proposed variants, we think the connection we make in this work between ConvRNNs and recent SSM methods can open up a new line of research to consider these alternative ConvSSM architectures. We hope this addresses the Reviewer's concerns. Please let us know if we can provide further clarification or information regarding ConvSSM variants and help to further increase the score.
> >
> > References: [1] Hungry Hungry Hippos: Towards Language Modeling with State Space Models. [2] Liquid Structural State-Space Models. [3] Efficiently Modeling Long Sequences with Structured State Spaces.

---

### Official Review · Reviewer_R5qx · 2023-07-04

**Soundness:** 2 fair
**Presentation:** 3 good
**Contribution:** 2 fair
**Rating:** 5
**Confidence:** 4

**Summary:**

The paper builds upon ConvRNNs and proposes the use of SSM (State Space Models) as a replacement for RNNs. This allows for efficient computation using parallel scan. The proposed method is evaluated on the Long Horizon Moving-MNIST Generation and Long-range 3D Environment Benchmarks datasets, where it achieves promising results.

**Strengths:**

The organization of this paper is clear and easy to understand. It starts from ConvRNN and replaces RNN with SSM, leading to the instantiation of ConvSSM known as ConvS5.

**Weaknesses:**

I think the main issue with this paper is the lack of motivation for the proposed approach, as replacing the matrix multiplication of SSM with convolution seems trivial.

**Questions:**

1. The motivation for the proposed approach should be made more explicit, whether it is based on intuitive explanations or considerations of efficiency, among other factors.

2. The experiments conducted in the paper are not comprehensive enough. It would be beneficial to compare the proposed method with other efficient attention-based approaches, such as kernel-based linear attention, 1+elu, performer, and cosformer, on long sequence tasks.

3. More ablation studies are needed to validate the rationale behind the design, specifically regarding the convolutional aspect. The existing ablation studies primarily focus on validating the initialization, but it is equally important to investigate and verify the effectiveness of the convolutional components.

References:
[1] Fast Autoregressive Transformers with Linear Attention
[2] Rethinking Attention with Performers
[3] cosFormer: Rethinking Softmax in Attention

**Limitations:**

Yes.

---

> ### Author Rebuttal · Authors · 2023-08-09
>
> We thank the reviewer for taking the time to review our paper and provide feedback.
>
> --**Re. motivation:** Thank you for this feedback. We were a little surprised since several other reviewers found the design well-motivated and clear (e.g. see Reviewer Kz49). Here we will walk through the motivation as presented in the paper. Please let us know where we can improve the presentation or provide further clarification.
>
> Spatiotemporal prediction methods need to scale efficiently with sequence length and effectively capture long-range dependencies in the spatiotemporal data. ConvRNNs have been historically popular due to their spatiotemporal modeling strengths and fast inference abilities. However, they are slow to train and suffer from vanishing/exploding gradients. On the other hand, Transformer methods provide strong performance but poor scaling in sequence length and slow inference. The recent SSM methods show subquadratic scaling in sequence length, ability to capture long-range temporal dependencies, and fast inference. Our method combines the best of ConvRNNs (strong spatiotemporal modeling abilities and fast inference) and SSMs (parallelizable and favorable scaling, long-range dependencies, and fast inference).
>
> We hope this answer provides clarification, but if the reviewer has concrete suggestions for how we can further improve this exposition of the motivation we would love to incorporate their recommended changes!
>
> --**Re. contributions**:
> We also note here that the move from SSMs to an effective version of a ConvSSM is not as trivial as simply replacing the SSM matrix-vector multiplications. Please see our detailed response regarding Technical Contributions in the General Response above where we outline the technical contributions required to provide a modern and scalable method.
>
>
> --**Re. experiments**: We were surprised by the feedback that our experiments were not comprehensive enough. We chose the most challenging and well-thought out long-range video prediction benchmarks that exist (the 3D Environment Benchmarks from TECO) and compare to state-of-the-art methods, many of which have been specifically designed for long-range video prediction.  We would like to note that reviewers Kz49 and xT6A complimented our paper for the broad benchmarking, comprehensive experiments and strong results of our paper and method. But we are also happy to include any additional baselines.
>
> As requested, we add the Performer baseline on DMLAB below. We see that ConvS5 also outperforms the Performer. We also note that one of our original baseline methods, Perceiver AR is a modern efficient attention alternative that was published more recently than linear attention or Performer and concurrently with Cosformer. We also include S5 as a baseline, which has been shown to significantly outperform Linear attention, Performer, CosFormer and a host of other efficient attention alternatives on long-range sequence tasks (see Table 10 in [1] and Table F.1 in [2] and also compare these to the results in Table 4 of the Cosformer paper [3].).
>
> **New Performer baseline on DMLab:**
> |             | FVD &darr; | PSNR &uarr; | SSIM &uarr; | LPIPS &darr; |
> |-------------|------------|-------------|-------------|--------------|
> | Perceiver-AR |  96     |      11.2      |    0.304     |    0.487       |
> | Performer   |     78     |     17.3   |    0.513   |      0.203    |
> | Transformer |     97     |     19.9    |    0.619    |     0.123    |
> | S5          |     221    |     19.3    |    0.641    |     0.162    |
> | ConvS5      |   **53**   |   **23.6**  |  **0.782**  |   **0.074**  |
>
> We are also in the process of training Performer on the Moving MNIST experiments and will add this baseline to the final version as well.
>
> --**Re. ablations**: While we did not explicitly label it as an ablation, the S5 baselines also serve as the best ablation of the ConvS5 convolutional approach, as it is the closest possible variant that does not include ConvS5's convolutions. We have also added an ablation of the nolinearity choice that connects the ConvS5 layers. Please see the new ablation table and the discussion in the General Response. We will add this separate Ablation Table and discussion to make the ablations more clear.
>
>
> **We thank the reviewer again** for taking the time to review our paper. We hope we have addressed the reviewer's questions and the reviewer is willing to increase their score. Please let us know if we can provide additional clarification or information.
>
> **References:** [1] Efficiently Modeling Long Sequences with Structured State Spaces. [2] Simplified State Space Layers for Sequence Modeling. [3] CosFormer: Rethinking Softmax in Attention.

---

> > ### Comment · Reviewer_R5qx · 2023-08-13
> > **Official Comment by Reviewer R5qx**
> >
> > Thanks for the author's response. Most of my questions have been addressed, and I am upgrading my rating from 4 to 5.

---

> > > ### Author Response · Authors · 2023-08-13
> > >
> > > We thank the reviewer for taking the time to read our rebuttal and for increasing our rating. Please let us know if we can address any additional questions/concerns to further increase the score.

---

### Official Review · Reviewer_Kz49 · 2023-07-10

**Soundness:** 3 good
**Presentation:** 4 excellent
**Contribution:** 2 fair
**Rating:** 6
**Confidence:** 4

**Summary:**

This paper proposes convolutional state space models (ConvSSMs), that combine ConvLSTM with the long sequence modeling approaches like S4/S5. In particular, authors propose ConvS5, that allow parallelizing the stateful autoregression of convRNNs across the sequential direction. Naively applying parallel scan speedup tricks like in S4/S5, leads to convolutional kernels of exploding kernel size. Hence, the authors propose an interesting way of structuring modeling complexity, choosing to have simple linear dynamics in the sequence direction that are learnt with 1x1 conv kernels, and complex non-linear operations in depth that can be learnt with entire resnet blocks. The authors benchmark the proposed ConvS5 model on the Moving-MNIST dataset by training on 300 and 600 frames and predicting 400/800/1200 frames conditioned on the initial 100 frames. Also benchmarking results are reported on the DMLab, Minecraft, and Habitat long-range benchmarks. Across the axes of image quality such as FVD, PSNR, SSIM and LPIPS as well as efficiency axes such as sampling speed.

**Strengths:**

+ **Well motivated and principled design** : The design considerations in ConvS5 is very well motivated with both theoretical (most of which are directly inspired from Deep SSMs) and practical arguments that make an interesting and worthwhile contribution to the community.

+ **Broad benchmarking and strong results** : The authors perform benchmarking across a number of datasets and benchmarks across all of which ConvS5 outperforms prior works such as ConvLSTM and a vanilla transformer on image quality as well as sample speed.

+ Paper is also well written with good exposition of the prior work ideas that are necessary to grasp convSSM which is difficult to balance given the large amount of history in S5 model family development.

**Weaknesses:**

- **Model Size** such as FLOPs / parameters and **Training Efficiency** such as Training Speed /  Number of Epochs need to reported for both convS5 and also all the baselines that are being compared to. Without these details, quality results on their own are meaningless since it is unclear if the best footing was also provided to the baselines as the proposed method. It is mentioned as an offhand remark that convS5 uses ResNet blocks as intermediate non linearities but that can lead to major slowdown and burdens (as also alluded to by the authors in limitations). If so, this should be made clear in the tables through the metrics mentioned before.

- **Boarder Ablations** : The paper is quite weak on ablation studies presented. Since the authors propose a new architecture, boarder ablations on more than just initialization are requires to be convincing of choices proposed.

- Experiments on non synthetic real data/video. While the authors do experiment on a bunch of datasets, they're all either toy like Moving-MNIST or, synthetic like DMLab/Minecraft/Habitat. For ConvS5 to be successful we need to make sure that it does well on real world benchmarks as well.


**Questions:**

- What hyperparameter sweeps and effort went into optimizing the Transformer performance on the proposed benchmarks? Was it commensurate with the proposed convS5? Also see weaknesses.

**Limitations:**

Yes

---

> ### Author Rebuttal · Authors · 2023-08-09
>
> We appreciate the time the reviewer took to review our paper and thank the reviewer for their positive feedback. We are glad the reviewer found the design of our method well-motivated and principled and also recognized the broad benchmarking of our paper and strong results of our approach.
>
> -- **Re. model size and training efficiency**: Please see the discussion of the computational cost and the other comparisons including flops between Transformer, ConvLSTM and ConvS5 in the General Response above. We also note that parameters, training speed, and max number of training steps were included for all experiments in Appendix C and D of the original submission. Parameter counts were included in the expanded Tables 5,7,8 and train step speeds were included in Tables 6,9. Parameters, max number of training steps and V100 days are listed in Tables 10-19. As the reviewer can see, the training footprints for the methods were commensurate. We will include a full summary of the tables in the final version to make this more clear.
>
> -- **Re. ablations:** Please see the discussion of ablations in the General Response above. Even though we did not explicitly label it as an ablation, the S5 baseline was meant to be an ablation on the convolutional structure of ConvS5, since it is the closest possible model without the convolutional structure. We have added a separate ablation table and discussion that makes the ablation of the convolutional structure (using S5) more clear and also added new ablations of the nonlinearity choice between layers.
>
> As our model sticks to a pretty basic architecture, there were not many other obvious design choices that can be ablated. If the reviewer has suggestions for other ablations we should run to improve the paper, please let us know and we will be happy to include!
>
> -- **Re. datasets:** We agree that real-world long-range spatiotemporal datasets are important to test and improve long-range spatiotemporal models. Unfortunately, strong benchmarks in this area do not currently exist, and the 3D Environment tasks from TECO are the most challenging and well-thought-out long-range video benchmarks we are aware of. We hope these recent works on long-range video prediction help to inspire the creation of better real-world long-range video benchmarks.
>
> --**Re. hyperparameter sweeps:** All of this information is included in Appendix D of the supplement of the original submission.
> Here is the summary:
>
> - Moving-Mnist: For the Transformer, we swept over 2 model sizes (hidden dimension of 512 and 1024) and 3 learning rates and chose the best model. For ConvS5 and ConvLSTM, we chose a single model size (less than the Transformer) and swept over the same learning rates as the Transformer.
>
> - DMLab: For each of the methods we chose model sizes and hyperparameters very close to those used in the TECO paper and swept over 3 learning rates for the Transformer, S5, ConvS5, TECO-Transformer, TECO-S5, and TECO-ConvS5, and chose the best run with the best learning rate for each model.
>
> - Minecraft: TECO-ConvS5 was run with two different learning rates with no other tuning due to the cost of this experiment.
>
> - Habitat: We only did 1 run with no further tuning due to the cost of this experiment.
>
> **Thank you again for your review.** We hope we have addressed the reviewer's questions and they are willing to increase their score. Please let us know if we can provide any additional clarification or information.

---

### Official Review · Reviewer_P1WH · 2023-07-11

**Soundness:** 2 fair
**Presentation:** 3 good
**Contribution:** 2 fair
**Rating:** 4
**Confidence:** 2

**Summary:**

This paper introduces a method for long-term sequential modeling. The authors extend prior RNN-based work for sequential modeling, i.e. S5[20],  by substituting the linear operations with convolutions.   The authors show the superiority of their method on the future prediction task with transformers. A favorable property of the RNN methods, in general, compared to transformers is that they scale linearly with respect to the sequence length, while the transformer's complexity is quadratic with respect to time.

**Strengths:**

1) The proposed method makes sense and is a natural extension of prior work.
2) The paper is well-written and easy to follow.

**Weaknesses:**

1) Compared to the transformers, the linear complexity with respect to the sequence length is clearly favorable. However, it would be informative to compare the actual computational cost, e.g. in terms of flops.

2) Despite transformers, RNNs are notorious for being difficult to train. It would be great to compare the training of two architecture as well. This can be a potential blocker for scaling the method for more complex datasets.

3) I am not an expert in this domain to have a proper evaluation on the impact of the experiments, but the contributions of this paper sound marginal with the NeurIPS standards. Also, it looks like the paper is missing some prior work to compare[1].

[1] Gao et al,  Simvp: Simpler yet better video prediction, CVPR 2022.

**Questions:**

Please see the weaknesses.

**Limitations:**

No, they have not. However, I do not see a particular negative social impact associated for this work.

---

> ### Author Rebuttal · Authors · 2023-08-09
>
> We thank the reviewer for the time they took to review our paper and for their feedback. We are glad the reviewer found the proposed method easy to follow.
>
> -- **Re. flops:** Please see the flops comparison in the computational cost section of the General Response above. Note the appendix of the original submission included the computational cost including training and inference runtimes, parameter counts, etc.
>
> -- **Re. training RNNs:** We agree RNNs are notoriously difficult to train, generally due to the vanishing/exploding gradient problem. We note the SSM line of work (e.g. S4/S5, etc.) that ConvS5 extends to the spatiotemporal domain effectively mitigates the vanishing/exploding gradient problem through its special parameterizations and initialization schemes. One of the major contributions of our work is to ensure the convolutional recurrence formulation retains these same favorable properties (See Section 3.3). We will add further discussion in the paper to make this more clear.
>
> In practice, we did not observe any stability issues when training ConvS5. However, we found that Transformer and ConvLSTM can be unstable with higher learning rates during the hyperparameter search (discussed in Appendix D), especially with Moving-MNIST trained on 600-length context. In addition, random initialization of the ConvSSM kernel instead of careful design of the kernel parameterization/initialization proposed in our paper (Section 3.3) was also unstable with higher learning rates.
>
> We do not think the recurrent structure of ConvS5 is a blocker to scaling the method, and the empirical results on complex, large scale datasets also support this. Please let us know if you have any other suggestions for comparing the training of the two architectures.
>
> -- **Re. contribution:** We respectfully disagree with the reviewer that the contributions are marginal. Please see the Technical Contributions section of the General Response above for a detailed explanation of the contributions.
>
> In short, we have introduced a method that can be parallelized during training like a transformer, provides fast autoregressive generation like an RNN, and provides high performance on complex video generation tasks requiring long-range reasoning and high-quality frames. Achieving these three aspects required contributions to ensure both computational efficiency and high performance.
>
> We show how convolutional recurrences can be parallelized both theoretically and practically (Section 3.2). To our knowledge, this has not been previously considered. Also, to capture long-range spatiotemporal dependencies, we developed a connection between the dynamics of SSMs and ConvSSMs (Proposition 3), which informs our parameterization and initialization design as discussed in Sections 3.3 and 3.4.
>
> Finally, our empirical results validate these design choices.
>
> -- **Re. SimVP:** We are happy to include a citation of SimVP in the related works. However, this work did not consider long-term prediction tasks (they only considered up to 40 frames). Since the SimVP model considers the sequence length as an input channel of the convolution performed by its Translator module, the SimVP model size has to scale with the sequence length. This may affect the scaling of this model to long-sequence video modeling. In our work, we compare against numerous state-of-the-art video prediction baselines designed for modeling long video sequences and evaluate on the most challenging existing video prediction baselines (training on hundreds of frames and predicting hundreds to thousands of frames). Nonetheless, we are working to include this method as an additional baseline. There was not time to complete the training and evaluation of this prior to the rebuttal deadline, but we will include it in the final version.
>
> **Thank you again** for taking the time to review our paper. We hope we have addressed the reviewer's questions and the reviewer is willing to raise their score. Please let us know if we can provide any additional clarification or information.

---

> > ### Comment · Reviewer_P1WH · 2023-08-20
> > **response to the authors**
> >
> > I thank the reviewers for their rebuttal, specifically for the comparisons in terms of FLOPs. However, I still believe the technical contributions of this paper do not match NeurIPS standards. I'll keep my rating as it is.

---

### Author Rebuttal · Authors · 2023-08-09

# General Response
We thank the reviewers for reviewing our submission and providing constructive feedback. We provide a general response here and respond to each reviewer individually. We presented the ConvS5 spatiotemporal sequence model which has parallelizable training, fast autoregressive inference, and effectively captures long-range spatiotemporal dependencies.

We were pleased all reviewers agree that the paper is **1)** easy to follow. Also, reviewers Kz49, rhqo, and xT6A appreciated  **2)** the effective model design for fast autoregressive generation and the parallelization process, and **3)** broad benchmarking and strong empirical results.

## Technical Contribution
There was some concern from reviewers P1WH and xT6A regarding technical novelty (though some reviewers such as Kz49 praised ConvS5's well motivated and principled design based on theoretical and practical arguments). We agree that the basic idea of going from SSM to ConvSSM by replacing the SSM's matrix-vector multiplications with convolutions is relatively straightforward. However, there are challenges to make this approach scalable and effective for modeling long-range spatiotemporal data.

**1. Computational efficiency, parallelization across the sequence for fast training and inference.** For the linear recurrence of SSMs, there are different ways to parallelize the model across the sequence (e.g. FFTs or parallel scans). However, a parallel scan for convolutional recurrences has not been studied before. In this paper, we are the first to introduce the parallelization of convolutional recurrences using a binary associative operator. In Section 3.2, we show both theoretical (Proposition 1) and practical results required to make this feasible and efficient.

**2. Capture long-range spatiotemporal dependencies.** We developed a theoretical connection between the dynamics of SSMs and ConvSSMs (Proposition 3, Section 3.3). Based on this result, in Section 3.4, we are able to introduce a parameterization and initialization design that allows ConvS5's convolutional recurrence to capture long-range spatiotemporal dependencies.

**Result:** The result of these contributions is ConvS5 which is parallelizable and overcomes difficulties during training (e.g., vanishing/exploding gradient problems) that traditional ConvRNN approaches experience. It also provides fast (constant time and memory per step) autoregressive generation compared to Transformers. Furthermore, our empirical results validate these design choices for modeling long-range spatiotemporal dependencies.

We argue that these contributions are not trivial, and our paper provides a rigorous framework that ensures both computational efficiency and modeling performance for spatiotemporal sequence modeling.

## Computational Cost
The appendix of the paper includes the experimental details including the number of parameters, training and inference times for all experiments (Tables 5-19). As a couple of reviewers asked us to provide FLOPs, we provide the comparison table below for Moving-MNIST trained on 600 frames. We will include the full summary in the final paper.

Although ConvS5 requires a few more FLOPs due to the convolution computations and some architecture choices (ResNet blocks for nonlinearity), our model is parallelizable during training (unlike ConvLSTM) and has fast autoregressive generation (unlike Transformer) ---  training 3x faster than ConvLSTM and generating samples 400x faster than Transformers. Note that the number of parameters are comparable.

|             | GFLOPS &darr; | Parallelizable     | Train Step Time (s) &darr; | Train cost (V100 days) &darr; | Sampling Speed (frames/s) &uarr; |
|-------------|:------------:|:------------:|:--------------:|:---------------------------:|:---------------------------------:|
| Transformer |      70.0     |       o      |         0.77 (1.0x)         |    50       |            0.21 (1.0x)            |
| ConvLSTM    |      64.9     |       x       |          3.0 (3.9x)       |    150      |             117 (557x)            |
| ConvS5      |      96.8     |       o      |         0.93 (1.2x)         |    50       |             90 (429x)             |

## Ablations
A common request was the ablation of the convolutional structure of ConvS5. We note that the S5 baseline also serves as an ablation of ConvS5 since it is the closest possible model without ConvS5's convolutional structure, even though we did not explicitly label this as an ablation. We will add a separate ablation table in the final version and make this more clear.

We additionally include the ablations of the nonlinear connections used between the ConvS5 layers. We use ResNet blocks but also consider elementwise GELU and GLU activations (used in the S5 paper). Due to the time limit, we only show the nonlinearity ablation for non-TECO models here. We will include the rest in the final version.

**DMLAB Ablations**
|                    | conv. | nonlinearity | FVD &darr; | PSNR &uarr; | SSIM &uarr; | LPIPS &darr; |
|--------------------|---|----------|------------|-------------|-------------|--------------|
| S5                 | x |   Glu    |     221    |     19.3    |    0.641    |     0.162    |
| ConvS5             | o | Gelu     |     129   |     21.4    |    0.709    |     0.110    |
| ConvS5             | o | GLU      |     112    |     21.6    |    0.720    |     0.098    |
| ConvS5 (ours)      | o | ResNet   |     53     |     23.6    |    0.782    |     0.074    |
| TECO-S5            | x |  Glu    |     35     |     20.1    |    0.687    |     0.143    |
| TECO-ConvSSM (random init.)      | o |    ResNet      |     44     |     21.0    |    0.691    |     0.010    |
| TECO-ConvS5 (ours) |  o |     ResNet    |     31     |     23.8    |    0.803    |     0.085    |

**We once again thank the reviewers for their time and positive and constructive feedback**. We will now respond to individual comments.

--The ConvS5 authors

---

> ### Author Response · Authors · 2023-08-18
>
> We thank the reviewers for the productive rebuttal discussion. To address questions from Reviewer xT6A regarding the convolutional structure, we performed additional ablations on the kernel sizes of ConvS5.  Since we think the additional experiments are also relevant for Reviewer Kz49 and Reviewer R5qx's requests regarding additional ablations, we move the answer we provided to Reviewer xT6A to the general response also:
>
> We would like to further address the reviewer's question: is there a large gain in $B$ and $C$ kernels being $k \times k$ instead of $1\times1$?
>
> The table below includes the ablation with different kernel sizes and nonlinearities on DMLAB. We ensure similar parameter counts for all models. These models have not fully converged yet (they are about halfway through and we evaluate all models at the same training step), but the results indicate that a larger kernel size significantly helps to improve the performance, especially for the FVD. We note that the rows with $1\times1$ kernels for both B and C are equivalent to vmapping an S5 layer which the reviewer mentioned. We will include the final results in the final version of the paper. These results reflect our observations during previous pilot experiments, though we did not fully quantify the gain. We thank the reviewer for highlighting this point as these results clearly improve the paper.
>
> **Kernel size ablations on DMLAB**
>
> |         B Kernel        |         C Kernel        | Nonlinearity | FVD &darr; | PSNR &uarr; | SSIM &uarr; | LPIPS &darr; |
> |:-----------------------:|:-----------------------:|:------------:|:----------:|:-----------:|:-----------:|:------------:|
> | $\boldsymbol{3\times3}$ | $\boldsymbol{3\times3}$ |  **ResNet**  |  **77\***  |  **22.3\*** | **0.740\*** |  **0.090\*** |
> |        $1\times1$       |        $3\times3$       |    ResNet    |     104    |     21.9    |    0.729    |     0.092    |
> |        $1\times1$       |        $1\times1$       |    ResNet    |     207    |     21.9    |    0.726    |     0.094    |
> |        $3\times3$       |        $3\times3$       |      GLU     |     122    |     20.7    |    0.677    |     0.118    |
> |        $1\times1$       |        $5\times5$       |      GLU     |     144    |     20.4    |    0.670    |     0.126    |
> |        $1\times1$       |        $1\times1$       |      GLU     |     381    |    19.95    |    0.644    |     0.134    |
>
> **bold**: final model, **\***: best number

---

### Comment · Area_Chair_Ppas · 2023-08-21
**Final summary on the originality and significance of your work**

Dear authors,

Thank you for submitting your work to NeurIPS, responding to the initial reviews provided by the reviewers, engaging in subsequent discussions with them and even conducting further experiments during this author-reviewer discussion phase.

As you can see, although most reviewers acknowledge the merits of your work, there is still reservation on recommending for paper acceptance due to various reasons even after further deliberation. As such, may I invite you to give a final summary in light of the further discussions with the reviewers? In particular, please address specifically the originality and significance of this work to justify its acceptance for presentation to the NeurIPS community. Originality and significance, among others, are major review criteria for acceptance, as you can see from the NeurIPS reviewer guidelines: https://nips.cc/Conferences/2023/ReviewerGuidelines.

We look forward to your summary (or “pitch”) soon.

Best,
AC

---

> ### Author Response · Authors · 2023-08-21
>
> As requested by the AC, below we provide a summary motivated by the discussion period focused on the originality and significance of our work.
>
> ### Summary
>  We presented the ConvS5 spatiotemporal sequence model. We introduced the idea of ConvSSMs, models with convolutional inductive biases of ConvRNNs, and linear, continuous-time dynamics as in SSMs such as S5. However, to address ConvRNN weaknesses (slow training and vanishing/exploding gradients), we addressed two technical challenges:
>
> **1. How to parallelize training across the sequence:**
>
> We are the first to introduce the parallelization of convolutional recurrences. In Section 3.2, we show both theoretical (Proposition 1) and practical results required to make this efficient and feasible.
>
> **2. How to effectively model long-range spatiotemporal dependencies:**
>
> To ensure ConvSSMs can capture long-range spatiotemporal dependencies, we developed a theoretical connection between the dynamics of SSMs and ConvSSMs (Proposition 3, Section 3.3). Based on this result, in Section 3.4, we are able to introduce a parameterization and initialization design that allows a ConvSSM's convolutional recurrence to capture long-range spatiotemporal dependencies.
>
> **Resulting Method**:  The methodological result is ConvS5 which is parallelizable and overcomes difficulties during training on long sequences that traditional ConvRNNs experience. It also provides fast autoregressive generation compared to Transformers. Our experimental results, including new ablations on the convolutional structure suggested by reviewers, support the design decisions and illustrate the effectiveness of the method compared to a number of strong baselines.
>
> We now address the Originality and Significance dimensions described in the Reviewer Guidelines linked above:
>
> ### Originality:
> *Are the tasks or methods new? Is the work a novel combination of well-known techniques? (This can be valuable!) Is it clear how this work differs from previous contributions? Is related work adequately cited*
>
> ConvS5 is a new method for spatiotemporal modeling that retains the favorable **convolutional inductive biases** and **fast inference** abilities of ConvRNNs, but unlike ConvRNNs, also has **parallelizable training** and can effectively capture **long-range spatiotemporal dependencies**.
>
> While ConvS5 can be viewed as a novel combination of the ideas of well-known ConvRNN methods and the relatively new (but increasingly popular) SSM methods, it required novel techniques to combine them in a practical and effective way. We propose a **novel parallel scan** for convolutional recurrences and introduce a **parameterization scheme** that retains the favorable dynamical properties of recent deep SSMs. The theory and discussions in Section 3.2 and 3.3 (and outlined above) provide a rigorous framework for how to think about these choices.
>
> We emphasize that ConvS5 is **not** a simple extension of SSMs such as S5.  ConvS5 is different than simply replacing a Transformer with an S5 model that acts on a sequence of spatial tokens. We show experimentally that ConvS5 outperforms this S5 baseline in spatiotemporal modeling. More specifically, we show a particular **convolutional structure choice and kernel sizes**  (including ablations suggested by xT6A, R5qx, and rhqo) are necessary to obtain the best performance.
>
> ### Significance:
>  *Are the results important? Are others (researchers or practitioners) likely to use the ideas or build on them? Does the submission address a difficult task in a better way than previous work? Does it advance the state of the art in a demonstrable way? Does it provide unique data, unique conclusions about existing data, or a unique theoretical or experimental approach?*
>
> Spatiotemporal modeling is an important and impactful domain. Unlike ConvRNNs, ConvS5 has **parallelizable training** and effectively captures **long-range dependencies**. Unlike Transformers, ConvS5 supports **fast autoregressive generation** and **unbounded context**. ConvS5 shows either state-of-the-art or comparable performance to state-of-the-art baselines on the most challenging long-range video prediction benchmarks available. We compare to a host of baseline methods including additional advanced RNN and efficient transformer baselines requested by reviewers xT6A and R5qx.
>
> Besides the method itself, an important contribution is developing the connection of ConvRNNs with recent SSM methods. In addition, during the rebuttal we proposed **additional ConvSSM variants** (ConvS4, H3 and Liquid ConvSSM variants), as suggested by Reviewer rhqo, to further connect ConvRNNs/ConvSSMs with recent advances in sequence modeling. More broadly, the connections and framework developed in this paper can open up a new line of research into these types of alternative, efficient spatiotemporal architectures.
>
> **We continue to welcome** concrete feedback from reviewers on these points to further improve the paper.

---

### Decision · Program_Chairs · 2023-09-21

**Decision:**

Accept (poster)

**Comment:**

This paper proposes a convolutional state space model that aims to overcome the efficiency concerns of previous methods, including ConvLSTM and transformers, for long-range spatiotemporal modeling.

The paper is generally clear and well written. The motivation for this work is presented well and the proposed model is based on a principled design. Strong results from the experiments are reported, showing that the proposed method can achieve significant speedup. Many of the comments given by the five reviewers were related to the experiments, demanding more extensive experimentation. We thank the authors for their effort in conducting further experiments and answering the questions in detail. Their effort has led four of the five reviewers to upgrade their overall ratings.

While we lean towards acceptance of this paper, I highly recommend the authors to consider the comments and suggestions of the reviewers thoroughly in revising their paper to make it appeal better to the research community.